# Profiling of microglia nodules in multiple sclerosis reveals propensity for lesion formation

Aletta M. R. van den Bosch [1] ✉, Marlijn van der Poel [1], Nina L. Fransen [1], Maria C. J. Vincenten[1], Anneleen M. Bobeldijk [1], Aldo Jongejan [2], Hendrik J. Engelenburg[1], Perry D. Moerland [2], Joost Smolders [1,3], Inge Huitinga [1,4,6] ✉ & Jörg Hamann [1,5,6] ✉

Microglia nodules (HLA-DR⁺ cell clusters) are associated with brain pathology. In this post-mortem study, we investigated whether they represent the first stage of multiple sclerosis (MS) lesion formation. We show that microglia nodules are associated with more severe MS pathology. Compared to microglia nodules in stroke, those in MS show enhanced expression of genes previously found upregulated in MS lesions. Furthermore, genes associated with lipid metabolism, presence of T and B cells, production of immunoglobulins and cytokines, activation of the complement cascade, and metabolic stress are upregulated in microglia nodules in MS. Compared to stroke, they more frequently phagocytose oxidized phospholipids and possess a more tubular mitochondrial network. Strikingly, in MS, some microglia nodules encapsulate partially demyelinated axons. Taken together, we propose that activation of microglia nodules in MS by cytokines and immunoglobulins, together with phagocytosis of oxidized phospholipids, may lead to a microglia phenotype prone to MS lesion formation.

Multiple sclerosis (MS) is a chronic neuroinflammatory disease characterized by focal demyelination and axonal damage throughout the brain and spinal cord[1,2]. Microglia are innate phagocytic glia cells of the central nervous system (CNS) that play an essential role in brain homeostasis[3,4]. In MS pathology, they contribute to the HLA-DR⁺ cell fraction phagocytosing myelin fragments in active and mixed active/inactive (mixed) lesions[1,5]. Despite many studies focusing on the role of microglia in MS, their particular role in lesion initiation is not defined yet.

White matter (WM) in MS that is not lesioned is called the normal appearing white matter (NAWM). Already in 1989, a magnetic resonance imaging (MRI) study showed alterations in the NAWM in MS compared to healthy control WM[6], which later were in part attributed to focal microglial activation in the absence of clear demyelination[7]. Abnormalities detected by MRI in NAWM distant from WM lesions could furthermore not be attributed to axonal pathology[8]. More recently, abnormalities in the NAWM in MS seen on MRI were followed over time and shown to predict the likelihood of developing subsequent MS lesions[9]. Accordingly, we recently identified subtle transcriptional changes in microglia in MS NAWM using bulk RNA sequencing[10]. Top differentially expressed (DE) genes related to lipid metabolism and phagocytosis that we found were also upregulated in

[1]Neuroimmunology Research Group, Netherlands Institute for Neuroscience, Amsterdam, The Netherlands. [2]Department of Epidemiology and Data Science, Amsterdam Public Health Research Institute, Amsterdam University Medical Center, Amsterdam, The Netherlands. [3]MS Center ErasMS, Department of Neurology and Immunology, Erasmus Medical Center, Rotterdam, The Netherlands. [4]Swammerdam Institute for Life Sciences, University of Amsterdam, Amsterdam, The Netherlands. [5]Department of Experimental Immunology, Amsterdam Institute for Infection and Immunity, Amsterdam University Medical Center, Amsterdam, The Netherlands. [6]These authors contributed equally: Inge Huitinga, Jörg Hamann. ✉e-mail: a.v.d.bosch@nin.knaw.nl; i.huitinga@nin.knaw.nl; j.hamann@amsterdamumc.nl

active MS lesions, indicating early demyelination by microglia in NAWM. Since microglia adapt to local changes in the CNS[11–13], sub-populations with distinct cellular states may differentially contribute to MS pathology.

HLA-DR+ ramified microglia can accumulate and cluster in the NAWM forming small clusters of at least four up to 50 cells that are in contact with each other, which was described in relation to MS pathology for the first time in 1993[14,15]. These so-called microglia nodules are regularly considered to precede MS lesion formation[16–24]. They are found in early as well as advanced MS cases and persist throughout the disease course[25,26]. Moreover, they are associated with axons undergoing Wallerian degeneration[19] and with encapsulation of activated complement deposits[16,17]. Microglia nodules are engaged in phagocytosis[27] and express both pro- and anti-inflammatory cytokines, such as tumor necrosis factor (TNF), interleukin (IL)−1β, and IL-10[28,29]. Van Horssen and colleagues reported expression of nicotinamide adenine dinucleotide phosphate (NADPH) oxidases by microglia nodules, which promotes the production of radical oxygens that can contribute to axonal damage[29]. In sum, microglia nodules in MS express molecules involved in immune regulation and oxidative stress. However, microglia nodules are not restricted to MS, since these are also found in relation to Wallerian degeneration in brain donors with traumatic brain injury, ischemia, or stroke[19,24], where microglia nodules line up around complement-opsonized axons similar as in MS[16,17,24]. Therefore, to disclose MS-specific characteristics of microglial nodules and their possible contribution to MS lesion formation, we compared microglia nodules in MS with microglia nodules in stroke and with surrounding non-nodular WM in MS and stroke. Considering the age and progressive clinical disease course of the donors[30] as well as the frequency of occurrence of microglia nodules, resolution of the microglia nodule is more likely than progression into an MS lesion[18]. We hypothesize that in MS a subset of microglia nodules will initiate MS lesion formation. As microglia nodules in stroke are not involved in lesion formation, differences between microglia nodules in MS and in stroke may reveal mechanisms behind MS lesion formation.

We assessed the pathological and clinical relevance of presence of microglia nodules in the MS autopsy cohort of the Netherlands Brain Bank (NBB). Of microglia nodules in MS and stroke tissue, the frequency and size were quantified. Gene expression in microglia nodules and non-nodular WM in MS and stroke was compared by RNA sequencing of laser micro-dissected tissue, and genes of interest were validated by immunohistochemistry (IHC). Using IHC, we studied the presence of lysosomal oxidized phospholipids, the incidence of adjacent T and B cells, the activation of the complement cascade and formation of the membrane attack complex, the appearance of the mitochondrial network, and the demyelination of axons encapsulated by microglia nodules. Our gene and protein expression data show that microglia nodules in MS are different to those found in stroke. They indicate that part of the microglia nodules in MS show all characteristics of very small and possibly starting MS lesions. Moreover, we identify molecules and pathways that may halt the progression of microglia nodules in MS into inflammatory, demyelinating lesions.

## Results
### Microglia nodules in MS correlate with MS pathology
In the MS cohort of the NBB, we quantified the number of MS donors with and without microglia nodules. To determine the relevance of microglia nodules for MS pathology and clinical course, we compared the pathological and clinical characteristics of MS donors with and without microglia nodule in all blocks dissected. Out of 167 MS brain donors, 107 donors (64%) had microglia nodules present in at least one tissue block dissected, and 60 donors (36%) did not have any microglia nodules in any of the tissue blocks dissected. MS donors with microglia nodules present in at least one tissue block, compared to MS donors without microglia nodules, had a significantly higher number of

**Table 1 | Donor demographics of NBB-MS cohort with and without nodules**

|  | Nodules present (all) N = 107 | No nodules present (all) N = 60 | p-value |
|---|---|---|---|
| Age (years) | 63.81 (13.73) | 63.55 (12.66) | 0.90 |
| Sex | 77F, 30M | 30F, 30M | 0.01 |
| MS type | 2 RR, 30 PP, 62 SP, 13 NA | 6 RR, 20 PP, 28 SP, 6 NA | – |
| Disease duration | 28.70 (13.39) | 29.60 (12.68) | 0.83 |
| Severity score | 2.38 (0.72) | 2.37 (0.76) | 0.71 |
| Reactive site load (log) | 0.70 (0.75) | 0.32 (0.56) | 1.4e-4 |
| Lesion load (log) | 1.79 (0.97) | 1.15 (1.10) | 5.0e-4 |
| Proportion active lesions | 0.23 (0.22) | 0.16 (0.22) | 0.03 |
| Proportion mixed lesions | 0.30 (0.24) | 0.21 (0.25) | 0.09 |
| Proportion inactive lesions | 0.30 (0.24) | 0.32 (0.26) | 0.03 |
| Proportion remyelinated lesions | 0.30 (0.28) | 0.37 (0.29) | 0.02 |
| MMAS score | 0.35 (0.26) | 0.30 (0.29) | 0.15 |

Provided is the mean ± standard deviation. Significance was tested with a two-sided Student's t test for continuous and normally distributed variables, a one-way ANOVA test for non-normally distributed variables and a $\chi^2$ test for binomial data.
F female, M male, MMAS microglia/macrophage activity score NA not available, NBB Netherlands Brain Bank, PP primary progressive, PR primary relapsing, RR relapsing-remitting, SP secondary progressive.

reactive sites (log, with nodules: 0.70 ± 0.75, without nodules: 0.32 ± 0.56, $p = 6.0e-4$) and higher lesion load (log, with nodules: 1.79 ± 0.97, without nodules: 1.15 ± 1.10, $p = 4.5e-4$) in standard locations, a higher proportion of active lesions (with nodules: 0.23 ± 0.22, without nodules: 0.16 ± 0.22, $p = 0.03$) and a lower proportion of inactive lesions (with nodules: 0.30 ± 0.24, without nodules: 0.32 ± 0.26, $p = 0.03$) and remyelinated lesions (with nodules: 0.30 ± 0.28, without nodules: 0.37 ± 0.29, $p = 0.02$) compared to MS donors without microglia nodules present. There was no difference in proportion of mixed active/inactive lesions, microglia/macrophage activity score (MMAS), disease severity measured as time to expanded disease disability scale (EDSS) 6, or disease duration (Table 1).

### Microglia nodules in MS are more frequent than in stroke but are not different in size
Nodule frequency and size were quantified in (NA)WM tissue sections of MS ($n = 7$) and stroke donors ($n = 7$). Donor demographics are summarized in Table 2. We observed a significantly higher number of microglia nodules in MS NAWM tissue as compared to stroke WM (MS: 18.1 ± 15.5 per 100 mm², stroke: 4.8 ± 2.2 per 100 mm², $p = 0.03$), but the microglia nodules were not different in size ($p = 0.16$, Fig. 1a–c). Furthermore, in contrast to stroke WM tissue with microglia nodules, the majority of MS NAWM tissue with microglia nodules showed many HLA-DR+ ramified microglia throughout the whole tissue section.

### Gene expression analysis shows diversity between microglia nodules in MS and stroke
To compare gene expression of microglia nodules in MS with those in stroke and non-nodular (NA)WM, tissue was manually dissected using laser capture microscopy. PCA showed partial clustering of four groups distinguishing MS and stroke tissue in the first dimension and nodular and non-nodular (NA)WM tissue in the second dimension (Fig. 2a). Cell type deconvolution analysis of the gene expression data showed that microglia nodules in MS compared to MS non-nodular NAWM were characterized by a significantly higher proportion of

**Table 2 | Donor demographics of MS and stroke donors**

| | Age (years) | Sex | PMD (h:min) | pH of CSF | MS type | Disease duration (years) | Time to EDSS6 (years) | Time of stroke to death (months) |
|---|---|---|---|---|---|---|---|---|
| MS (n = 7) | 66 (12) | 5M/2F | 8:50 (7:19) | 6.47 (0.10) | 6 SP/1 PP | 38 (12) | 26 (15) | – |
| Stroke (n = 7) | 80 (10) | 4M/3F | 7:18 (4:38) | 6.42 (0.24) | – | – | – | 64 (111) |
| p-value | 0.06 | 0.58 | 0.43 | 0.59 | – | – | – | – |

Data presented as average ± standard deviation.
PMD post-mortem delay, CSF cerebrospinal fluid.

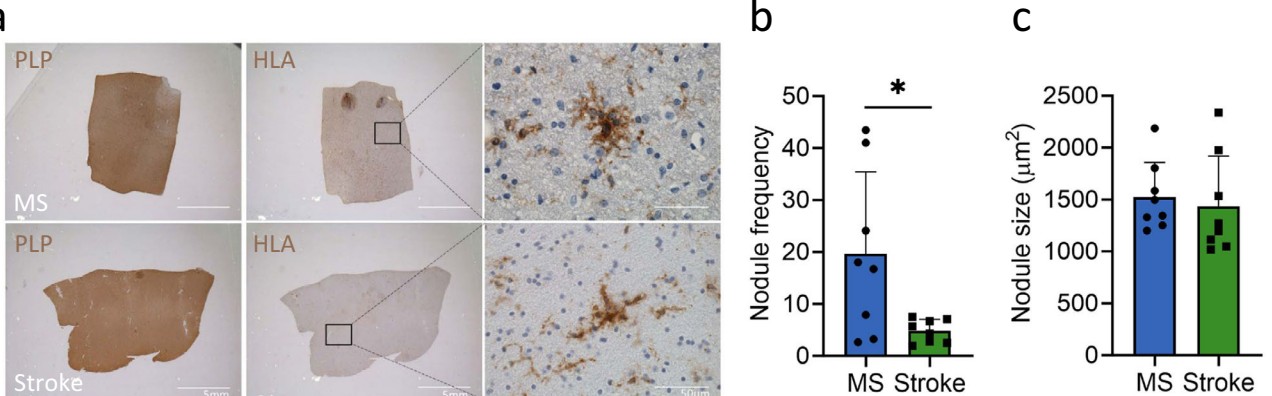

**Fig. 1 | Microglia nodules in MS are more frequent than in stroke but are similar in size.** IHC stainings with HLA and PLP shown in brown. Quantifications performed on n = 8 MS and n = 8 stroke donors. **a** PLP and HLA staining of (NA)WM matter in MS and in stroke shows no sign of demyelination and clustering of HLA-DR$^+$ cells into nodules. **b** In MS, 197 nodules were counted in total and in stroke 60 nodules were counted in total. Microglia nodule frequency was calculated per donor as number of microglia nodules per 100 mm$^2$. The nodule frequency was higher in MS compared to stroke (p = 0.03). **c** In MS, 197 nodules were measured in total, and in stroke, 46 nodules were counted in total. Microglia nodule size as measured in µm$^2$ was similar in MS and stroke. Bar plots show mean ± standard deviation. Significance was tested with a two-sided Student's t test, p value < 0.05 is indicated with *. Source data are provided as a Source Data file.

microglia cells (p = 0.03) and endothelial cells (p = 6.3e-4) and a lower proportion of oligodendrocytes (p = 7.7e-3). There were no significant differences in estimated cell type proportions between microglia nodules in MS and microglia nodules in stroke, microglia nodules in stroke and stroke non-nodular WM, or MS non-nodular NAWM and stroke non-nodular WM (Fig. 2b). As validated with IHC, microglia nodules were equally often in contact with vessels in MS and in stroke (Suppl Fig. 1). As our main comparison of interest was between microglia nodules in MS and stroke, we focused on the results of the analysis without correction for microglia proportion. Results of the analysis with correction for the proportion of microglia are shown in the text (hereafter 'after correction') and are also included in Suppl Tables 3 and 4. For each comparison, up to 50 top DE genes are shown in Suppl Table 3–6. Expression of all genes of interest mentioned below is summarized in Suppl. Table 7. The highest number of DE genes was observed in microglia nodules in MS vs MS non-nodular NAWM, where 325 DE genes were upregulated. In microglia nodules in MS vs microglia nodules in stroke, 256 DE genes were upregulated, of which 40 DE genes were also upregulated in microglia nodules in MS vs MS non-nodular NAWM and are considered MS nodule-specific genes. The lowest number of DE genes was found in microglia nodules in stroke vs stroke non-nodular WM with 10 upregulated DE genes. In MS non-nodular NAWM vs stroke non-nodular WM, 23 DE genes were upregulated (Fig. 2c). The number of downregulated DE genes was low in all comparisons. Microglia nodules compared to non-nodular (NA)WM in MS and in stroke shared only few communally upregulated DE genes (C1qB, RPGR, and SLC11A1), which are associated with involvement in activation of the classical complement pathway and in phagocytosis. After correction, C1qB remained significantly differentially expressed for both comparisons, while SLC11A1 was only significantly upregulated by microglia nodules in stroke compared to stroke non-nodular WM (Fig. 2c–e).

## Microglia nodules in MS but not in stroke express MS lesion pathology-associated genes

Microglia nodules compared to non-nodular NAWM in MS had a significantly higher expression of a multitude of genes previously associated with MS pathology (CXCL16, IL18, MX1, LPL, CD14, CD83, IL1B, CDKN1A, GPNMB, HLA-DRB5, C1QA, C1qB, SPP1, TLR6, CHI3L1, after correction CXCL16, IL18, C1QA, C1qB remained significant)[1,10,27,28,31–37]. C1qB was also upregulated by microglia nodules in stroke compared to stroke non-nodular WM, also after correction. Microglia nodules in MS, compared to microglia nodules in stroke, also upregulated expression of genes previously associated with MS-lesion pathology (HLA-DRB5, ISG15, MX1, after correction HLA-DRB5, ISG15) and MS susceptibility (IFNAR2, also after correction)[32,38]. In MS non-nodular NAWM compared to stroke non-nodular WM, no genes previously associated with MS pathology were differentially expressed (Fig. 2d–g). Thus, microglia nodules in MS and not in stroke show MS lesion-related activation. Markers for phagocytic exhaustion (CD22, NOS1, CYBB, CD68, and CYBA)[39] were all numerically but not significantly upregulated in microglia nodules in MS compared to stroke (Suppl Fig. 2).

## Microglia nodules in MS express genes indicative for lesion formation

Using the DAVID algorithm for functional annotations, we found DE genes in microglia nodules in MS compared to microglia nodules in stroke and compared to MS non-nodular NAWM that functionally may be indicative for lesion formation (Suppl Fig. 3 and Suppl. Tables 8 and 9). Microglia nodules in MS had upregulated genes that imply involvement in the adaptive and the innate immune response (compared to MS non-nodular NAWM: HLA-DMB, JAK3, TLR6, IFI27, none after correction; compared to microglia nodules in stroke: IDH1, PSME3, IFNAR2, ISG15, PARP9, IL33, after correction IFNAR2, ISG15, PARP9), phagocytosis (compared to MS non-nodular NAWM:

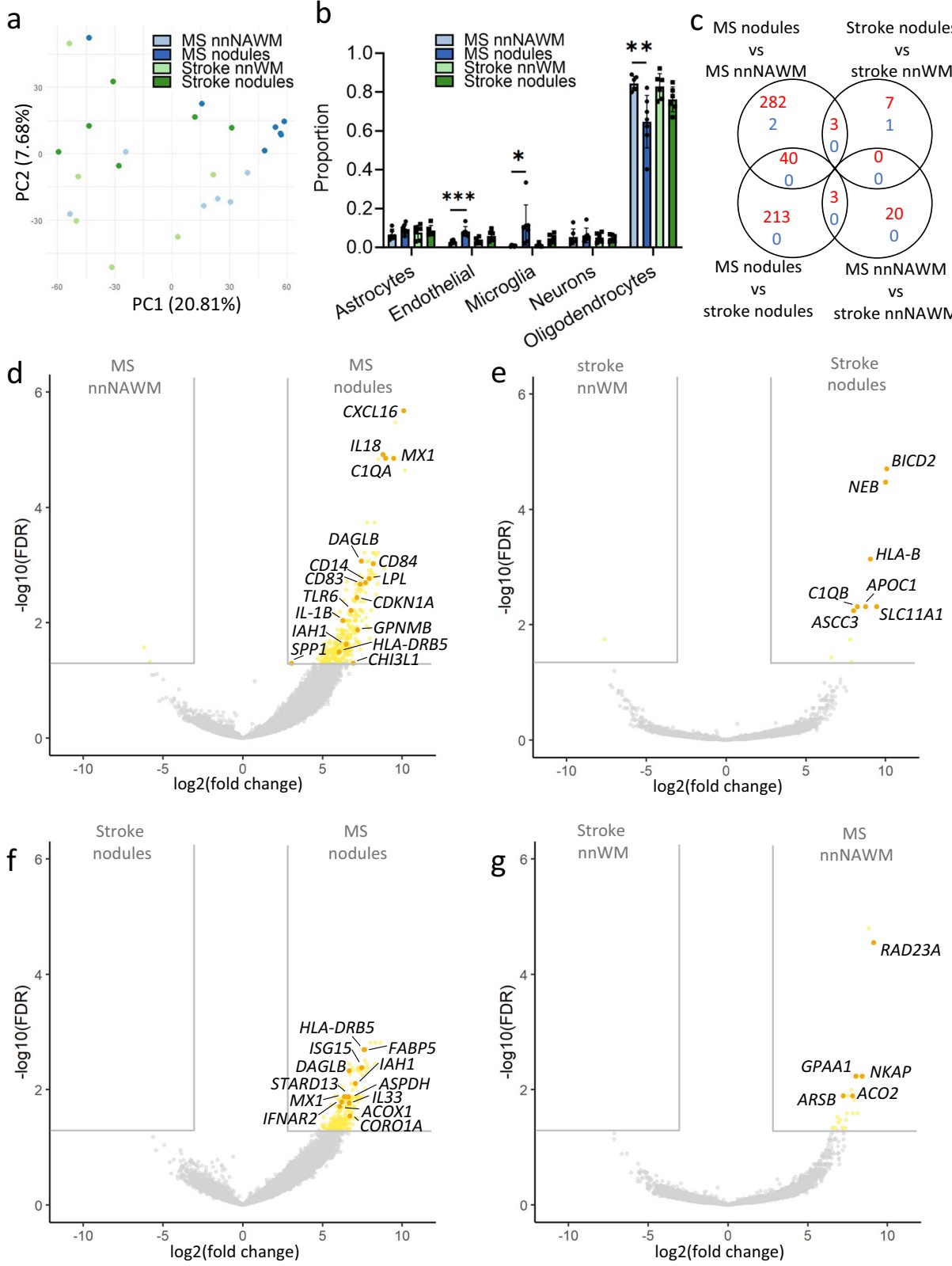

CD14, IRF8, MERTK, NCF2, none after correction), and lipid metabolic processes (compared to MS non-nodular NAWM: LPL, DAGLB, IAH1, after correction DAGLB; compared to microglia nodules in stroke: PLCD3, FABP5, ACLY, IAH1, DAGLB, CHI3L1, CHI3L2, STARD13, GPCPD1, after correction FABP5, IAH1, DAGLB, STARD13). Microglia nodules in MS had an increased expression of genes implying T- and B-cell homeostasis and proliferation (compared to MS non-nodular NAWM:

NCKAP1L, CASP3, JAK3, FADD, TCIRG1, after correction NCKAP1L; compared to microglia nodules in stroke: CORO1A, NCKAP1L, CASP3, APBB1IP, none after correction) and natural killer T (NKT) cell-mediated cytotoxicity (compared to microglia nodules in stroke: GRB2, CASP3, BID, none after correction). In microglia nodules in MS, genes indicating immunoglobulin (Ig) signaling were upregulated as compared to MS non-nodular NAWM: HLA-DMA, HLA-DPB1, HLA-

**Fig. 2 | Microglia nodules in MS and stroke share only few commonalities, and microglia nodules in MS show lesion-associated microglia activation.** Analysis performed on $n = 6$ MS NAWM samples, $n = 7$ MS nodule samples, $n = 6$ stroke NAWM samples, $n = 6$ stroke nodule samples. **a** PCA plot showing discrimination of MS and stroke samples on the first dimension and discrimination of nodules and non-nodular NAWM in the second component. **b** Proportion of cell types in the groups, shown as mean ± standard deviation. Statistical significance was tested with a two-sided Student's $t$ test. Microglia nodules in MS had a higher proportion of endothelial cells ($p = 6.3e-4$) and microglia cells ($p = 0.03$), and a lower proportion of oligodendrocytes ($p = 7.7e-3$) compared to MS NAWM. *$p$ value < 0.05, **$p$

value < 0.01, ***$p$ value < 0.001. **c** Venn diagram showing the number of DE genes with adjusted $p$-value < 0.05 and logFC > 3 or <−3, either upregulated in red or downregulated in blue between the various groups. Differential expression was assessed using an empirical Bayes moderated $t$-test within limma's linear model framework including the precision weights estimated by voom and the consensus correlation between samples of the same donor. Gene expression of (**d**) MS nodules versus non-nodular NAWM, (**e**) stroke nodules versus stroke non-nodular WM, (**f**) MS nodules versus stroke nodules, and (**g**) MS non-nodular NAWM versus stroke non-nodular WM. DE genes are highlighted in yellow, and top DE genes are highlighted in orange. Source data are provided as a Source Data file.

*DMB, HLA-DRB1, HLA-DRB5*, none after correction; compared to microglia nodules in stroke: *HLA-DRB5* (also after correction) and of cytokine signaling, specifically interferon (IFN)γ and TNF (compared to MS non-nodular NAWM: *IRF8, CD84, LRRK2, FADD, LPL, TLR2, EGR1, IL18*, after correction *IL18*; compared to microglia nodules in stroke: *IFNAR2*, also after correction). Furthermore, gene expression of microglia nodules in MS indicates they may be under metabolic stress (compared to MS non-nodular NAWM: *PPRC1, SAMM50, MTG1*, after correction *MTG1*; compared to microglia nodules in stroke: *ACOX1, COA4, MTG1, PPRC1, SAMM50*, after correction *MTG1*) and may be responding to as well as producing reactive oxygen species (ROS) (compared to microglia nodules in stroke: *ASPDH, BID, CRYZL1, IDH1, MAOB, SMAD3*, none after correction).

### Microglia nodules in MS reside in an inflammatory environment

As gene expression analysis revealed the likelihood of nearby lymphocytes, this was assessed using IHC. In MS and not in stroke, CD20+ B cells and CD138+ plasma cells were found in close proximity to microglia nodules. CD38+ plasmablasts were observed more frequently near MS compared to stroke microglia nodules (CD20: MS: 5 of 106 nodules, per donor 5.2% ± 10.4%, stroke: 0 out of 83 nodules, 0% ± 0% per donor; CD138: MS: 13 of 162 nodules, per donor 6.7% ± 10.7%, stroke: 0 of 78 nodules, per donor 0% ± 0%; CD38: MS: 90 of 202 nodules, per donor 34.6% ± 24.5%, stroke: 1 of 65 nodules, per donor 0.4% ± 0.9%, $p = 6.7e-3$, Fig. 3a–f). Some microglia nodules with lymphocytes in close proximity were in direct contact with the lymphocytes (CD20: 0%, CD138: 8%, CD38: 38%, CD3: 14%). For microglia nodules not in direct contact with the lymphocytes, the distance was for CD20+ B cells 81.52 ± 53.36 μm, for CD138+ plasma cells 71.48 ± 39.44 μm, for CD38+ plasmablasts 71.48 ± 39.44 μm, and for CD3+ T cells 50.47 ± 35 μm. In line with the presence of B and plasma cells in close proximity to MS but not near microglia nodules in stroke, with IHC, we found IgG deposition in some MS cases (4/7 MS donors) in the lumen of some vessels and IgG-producing plasmablasts (2/7 MS donors) near microglia nodules (Fig. 3g, h) and never in stroke tissue. Also, we found significantly upregulated expression of the Ig genes *IGKC, IGHG1, IGHG2*, and *IGKV3-15* in MS nodule NAWM tissue as compared to stroke (Suppl Fig. 4). Together, this indicates that Ig secretion only takes place in tissue containing microglia nodules in MS but not in stroke. In MS, CD3+ T cells were more frequently observed near microglia nodules compared to stroke (MS: 38 of 151 nodules, per donor 26.7% ± 9.1%, stroke: 5 of 89 nodules, per donor 5.1% ± 5.1%, $p = 0.02$, Fig. 3i, j). In MS, both CD4+ as well as CD8+ T cells were found in close proximity to microglia nodules in MS (CD4: per donor 8.3 ± 6.8 nodules, CD8: per donor 18.0 ± 16.5 nodules, Fig. 3k–m). In MS, subsets of microglia nodules and CD3+ T cells expressed the proliferation marker PCNA, suggesting that these T cells have encountered antigenic re-stimulation (Fig. 3n, o).

### Classical complement pathway activation in MS leads to MAC formation

Microglia nodules in MS as well as in stroke have a higher expression of *C1Q* genes compared to non-nodular (NA)WM, which is in line with a previous study showing complement deposition presence in both MS

and microglia nodules in stroke[24]. C1q is a complement component expressed by microglia and macrophages that can bind to the Fc tail of Igs[40] and is a critical mediator of microglia activation in MS[41]. As Ig-related genes were only expressed in MS tissue and not in stroke tissue, this may potentially lead to complete activation of the complement cascade causing cell lysis in MS but not in stroke. Therefore, we stained the microglia nodules in MS and in stroke for complement components C1qB, C3d, and the membrane attack complex (MAC). In MS and stroke, equal percentages of microglia nodules expressed C1qB (MS: 63 of 163 nodules, per donor 36.6% ± 6.0%, stroke: 22 of 68 nodules, per donor 30.5% ± 19.0%, $p = 0.25$, Fig. 4a–c). In MS, microglia nodules more often were associated with C3d+ axons compared to microglia nodules in stroke (MS: 35 of 150 nodules, per donor 17.2% ± 15.5%, stroke: 4 of 54 nodules, per donor 4.2% ± 7.0%, $p = 0.008$), suggesting an increased activation of the complement pathway in MS (Fig. 4d–f). Furthermore, microglia nodules in MS were more often associated with C5b-9, which constitutes the MAC, compared to microglia nodules in stroke (MS: 29 of 106 nodules, per donor 31.40% ± 17.37, stroke: 16 of 117 nodules, per donor 12.96% ± 7.54%, $p = 0.03$), indicating involvement of microglia nodules in MS in complement-mediated tissue degeneration (Fig. 4g–i).

### Increased lipid metabolism in microglia nodules in MS

Genes of interest involved in lipid metabolism that were upregulated in microglia nodules in MS compared to microglia nodules in stroke were validated with IHC. HLA+ microglia in the non-nodular (NA)WM and those surrounding microglia nodules (distance <100 μm, Suppl Fig. 5) rarely expressed fatty acid-binding protein 5 (FABP5), diaglycerol lipase-beta (DAGLB), StAR-related lipid transfer domain protein 13 (STARD13), or isoamyl acetate hydrolyzing esterase 1 (IAH1). The percentage of HLA+ cells that were also FABP5+, DAGLB+, STARD13 or IAH1+ was similar for MS and stroke (NA)WM. Microglia nodules in MS compared to microglia nodules in stroke significantly more often expressed FABP5 (MS: 81 of 143 nodules, per donor 55.2% ± 8.7%, stroke: 14 of 56 nodules, per donor 22.0% ± 19.6%, $p = 3.3^{-6}$), DAGLB (MS: 62 of 79 nodules, per donor 73.8% ± 10.0%, stroke: 12 of 35 nodules, per donor 19.2% ± 22.5%, $p = 1.11^{-8}$), and STARD13 (MS: 28 of 57 nodules, per donor 47.5% ± 19.9%, stroke: 7 of 24 nodules, per donor 25.1% ± 20.9%, $p < 0.05$). Difference in expression of isoamyl acetate hydrolyzing esterase 1 (IAH1) did not reach significance due to large variation (MS: 21 of 88 nodules, per donor 43.7% ± 31.0%, stroke: 7 of 24 nodules, per donor 25.1% ± 20.9%, $p = 0.32$, Fig. 5a–l).

In MS, the NAWM has more oxidized phospholipids compared to controls[42], which may be one of the triggers for microglia in MS to cluster and form nodules in order to clear up the oxidized phospholipids. Therefore, using IHC and Stimulated Emission Depletion (STED) microscopy, the percentage of microglia nodules that had phagocytosed oxidized phospholipids were quantified in MS and stroke. In microglia nodules in MS, lysosomal-associated membrane protein 1 (Lamp1)+ lysosomes more often contained oxidized phospholipids (detected using antibody E06) compared to microglia nodules in stroke (MS: 30 of 47 nodules, per donor 68.0% ± 2.8%, stroke: 7 of 29 nodules, per donor 29.1% ± 7.0%, $p = 3.2^{-12}$), showing that microglia

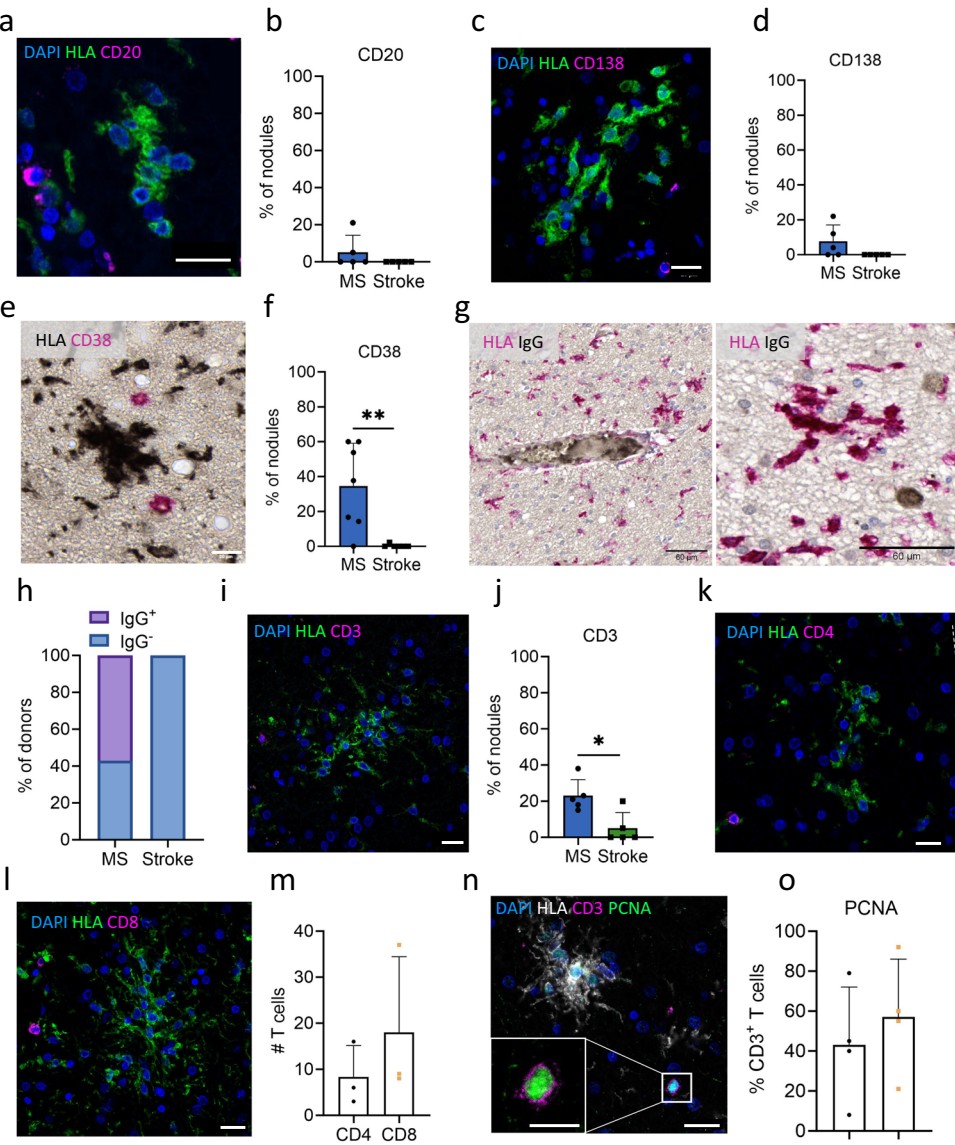

**Fig. 3 | Microglia nodules in MS reside in a more inflammatory environment compared to stroke nodules.** IHC stainings. **a, b** A CD20⁺ B cell (CD20 in magenta, HLA in green), quantified in $n = 5$ MS donors and $n = 5$ stroke donors, (**c, d**) a CD138⁺ plasma cell (CD138 in magenta, HLA in green), quantified in $n = 5$ MS donors and $n = 5$ stroke donors, and (**e, f**) two CD38⁺ plasmablasts (CD38 in magenta, HLA in black) in MS tissue, quantified in $n = 7$ MS donors and $n = 6$ stroke donors, which are seen not or less often in stroke, reaching significant difference for CD38 ($p = 6.7e-3$). **g, h** IHC of IgG (IgG in black HLA in magenta) in MS shows IgG staining in the lumen of a blood vessel in close proximity to a nodule and an IgG⁺ plasmablast in close proximity to a microglia nodule, which was not found in stroke donors. Quantified in $n = 7$ MS donors and $n = 7$ stroke donors. **i, j** A CD3⁺ T cell (CD3 in magenta, HLA in green) in MS tissue, which was less frequent in stroke, quantified in $n = 5$ MS donors and $n = 6$ stroke donors. **k–m** In MS, both CD4⁺ and CD8⁺ T cells (CD4 and CD8 in magenta, HLA in green) are found near microglia nodules in MS, quantified in $n = 3$ MS donors. **n, o** Activated PCNA⁺ CD3⁺ T cell (CD3 in magenta, PCNA in green, HLA in gray) in close proximity to microglia nodules in MS, quantified in $n = 4$ MS donors. Bar plots show mean percentage of nodules ± standard deviation or percentage of donors. Significance for proportional data was tested with a two-sided quasibinomial generalized linear model without correction for multiple testing or a Student's $t$ test for continuous numerical data, p value < 0.05 is indicated with *. White scale bars indicate 20 μm in overview images and 10 μm in zoomed images, black scale bars indicate 60 μm. Source data are provided as a Source Data file.

nodules in MS are more involved in phagocytosis of oxidized phospholipids (Fig. 6a-c).

As microglia nodules in MS are likely involved in lipid metabolism, we set out to investigate if microglia nodules are involved in demyelination with high-resolution IHC. Cryo-protected MS normal-appearing optic nerve tissue[43] provided sufficient resolution to investigate demyelination of individual axons encapsulated by microglia nodules. In MS but not in controls, some axons surrounded by microglia nodules showed partial demyelination, as indicated by loss of PLP staining of a part of the SMI312⁺ axon (MS: 9 of 20 nodules, per donor 41.0% ± 14.2%, controls: 0 of 24 nodules, 0.0% ± 0.0%, $p = 2.2^{-16}$, Fig. 6d, e).

## Mitochondrial network in microglia nodules in MS is more tubular

DE genes of microglia nodules in MS compared to microglia nodules in stroke were indicative of an altered metabolic state. Using IHC and STED microscopy, we assessed the mitochondrial network of microglia nodules in MS and in stroke, and we quantified the mitochondria frequency and size in axons encapsulated by microglia nodules in MS and in controls. Of each microglia nodule, the translocase of the outer mitochondrial membrane (TOMM20) network was classified as fragmented, intermediate, or tubular. Quantification of the mitochondrial network in microglia nodules in MS and in stroke showed that in MS the mitochondrial network was more often tubular (MS: 51.0% ± 18.5%,

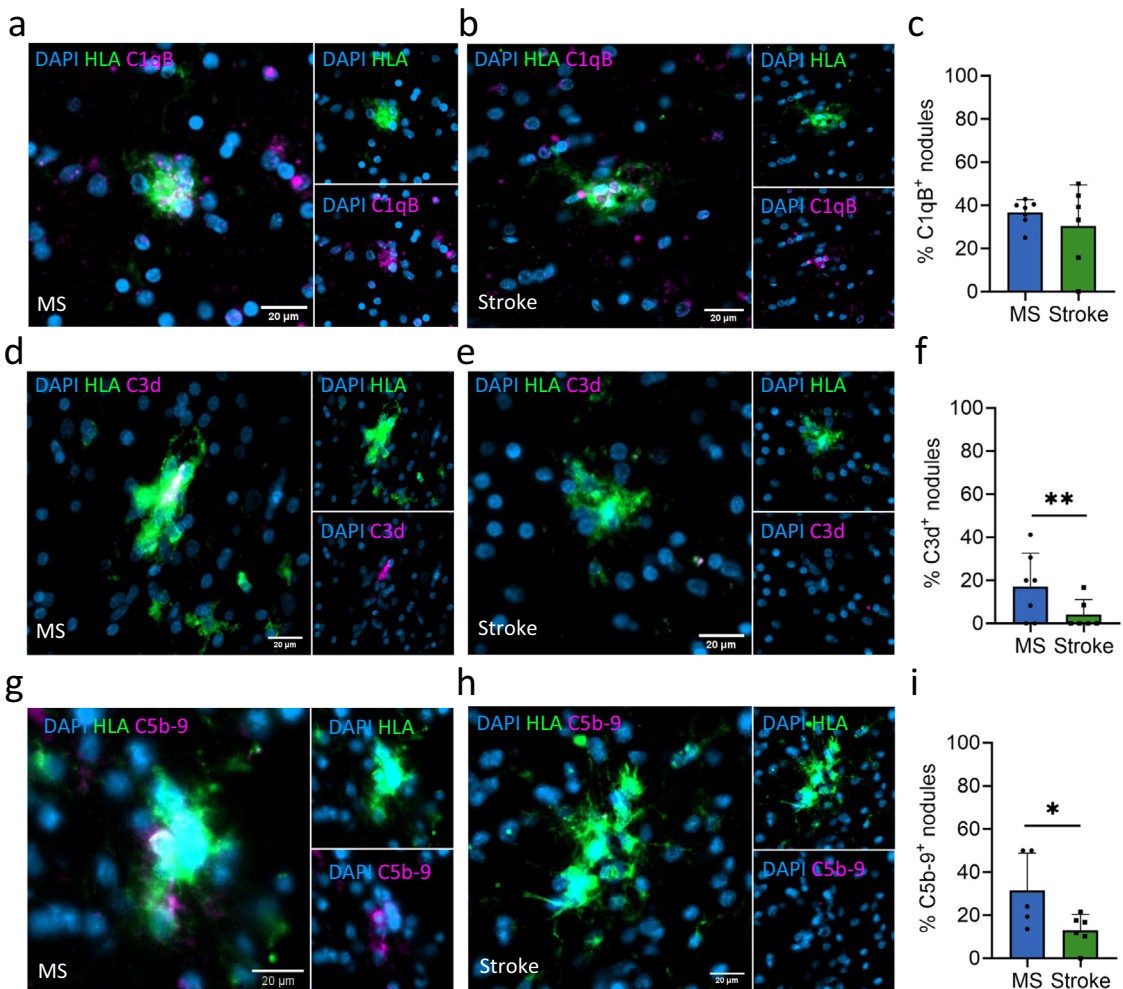

**Fig. 4 | Microglia nodules in MS are associated with activation of the classical complement pathway and are associated with membrane attack complex formation.** IHC stainings (C1qB, C3d and C5b9 in magenta, HLA in green). C1qB⁺ HLA⁺ microglia nodule (**a**) in MS and (**b**) in stroke. **c** Microglia nodules in MS and in stroke equally often express C1qB, quantified in $n = 7$ MS donors and $n = 6$ stroke donors. **d** HLA⁺ microglia nodule associated with a C3d⁺ axonal fragment in MS and (**e**) an HLA⁺ microglia nodule in stroke that is not associated with a C3d⁺ axonal fragment. **f** In MS, microglia nodules are more often associated with C3d⁺ axonal fragments than in stroke ($p = 0.008$), quantified in $n = 7$ MS donors and $n = 6$ stroke donors. IHC of (**g**) an HLA⁺ C5b-9⁺ microglia nodule in MS and (**h**) an HLA⁺ microglia nodule in stroke that is not C5b-9⁺. **i** In MS, microglia nodules are more often C5b-9⁺ than in stroke ($p = 0.03$), quantified in $n = 6$ MS donors and $n = 6$ stroke donors. Bar plots show mean value ± standard deviation. Significance was tested with a two-sided quasibinomial generalized linear model without correction for multiple testing, *$p$ value < 0.05, **$p$ value < 0.01. Source data are provided as a Source Data file.

Stroke: 13.1% ± 14.9%, $p = 0.009$) (Fig. 7a–c). This tubular network is indicative of a hypermetabolic state of the microglia nodules in MS. In axons encapsulated by microglia nodules, there was no difference in mitochondria frequency nor size.

## Discussion

Here, we have studied the potential involvement of microglia nodules in MS lesion formation by correlating their presence with pathological and clinical characteristics and by comparing microglia nodules in MS to microglia nodules in stroke and to surrounding non-nodular (NA) WM using RNA sequencing of laser-microdissection captured tissue and IHC. The combination of methods provided a highly detailed and comprehensive dissection of the microglia nodule in MS as putative precursors of lesion formation in MS. We show that MS microglia nodules (1) correlate with more severe MS pathology, (2) upregulate expression of genes similar to MS-lesions, (3) upregulate expression of genes associated with adaptive and innate immune responses, lymphocyte activation, phagocytosis, lipid metabolism, and metabolic stress, (4) have nearby lymphocyte presence, (5) contain partially demyelinated axons, (6) are associated with IgG transcription,

complement activation, and MAC formation, and (7) are in a hyper-metabolic state associated with increased pro-inflammatory cytokine and ROS gene expression. Our findings indicate that the nodule milieu may be held responsible for early lesion formation.

We here show that microglia nodules in MS are pathologically relevant. MS donors with microglia nodules compared to those without microglia nodules had a higher lesion load and reactive site load. The proportion of active lesions was higher, and the proportion of inactive and remyelinated lesions was lower in MS donors with microglia nodules compared to those without. Interestingly, there was no difference in the proportion of mixed active/inactive lesions. As we hypothesize that active lesions precede mixed active/inactive lesions, this indicates that microglia nodules are possibly associated with new MS lesion formation. Therefore, MS donors without microglia nodules may represent a subgroup in which frequency of new lesion formation has decreased, whereas those with microglia nodules may still develop new MS lesions. The presence of microglia nodules was not associated with clinically more severe MS. Although microglia nodules in MS do not seem to represent previous clinical progression, this does not exclude the possibility that they may represent future clinical

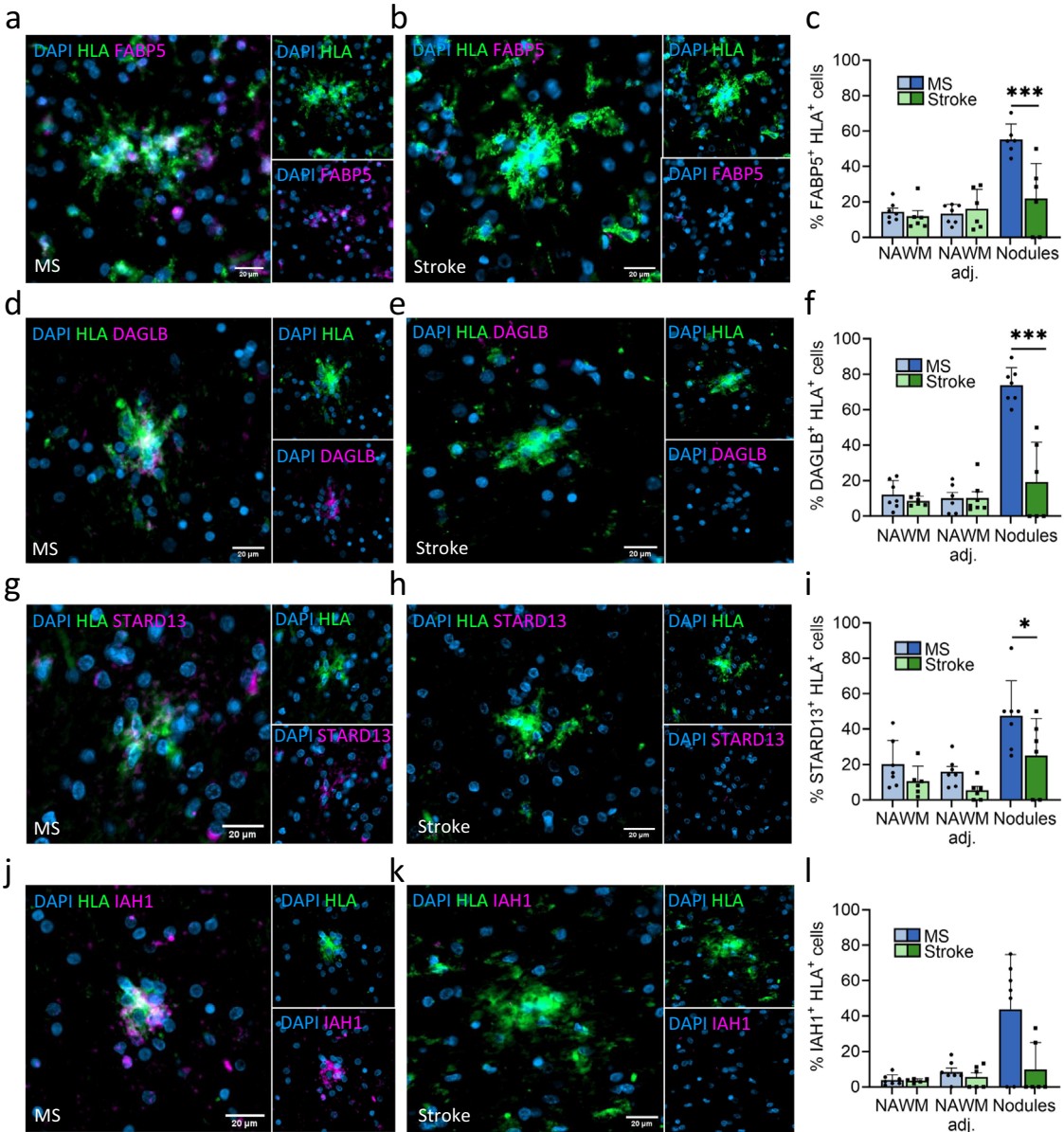

**Fig. 5 | Microglia nodules in MS are involved in lipid metabolism.** IHC stainings (FABP5, DAGLB, STARD13, and IAH1 in magenta, HLA in green). **a** An HLA⁺ FABP5⁺ microglia nodule in MS and (**b**) an HLA⁺ FABP5⁻ microglia nodule in stroke. **c** In MS, microglia nodules more often express FABP5 than in stroke ($p = 3.3e\text{-}6$). NAWM quantified in $n = 7$ MS donors and $n = 6$ stroke donors, NAWM adjacent to nodules quantified in $n = 7$ MS donors and $n = 6$ stroke donors, nodules quantified in $n = 6$ MS donors and $n = 6$ stroke donors. **d** An HLA⁺ DAGLB⁺ microglia nodule in MS and (**e**) HLA⁺ DAGLB⁻ microglia nodule in stroke. **f** In MS, microglia nodules more often express DAGLB compared to microglia nodules in stroke ($p = 1.11e\text{-}8$). NAWM quantified in $n = 6$ MS donors and $n = 7$ stroke donors, NAWM adjacent to nodules quantified in $n = 7$ MS donors and $n = 6$ stroke donors, nodules quantified in $n = 7$ MS donors and $n = 6$ stroke donors (**g**) An HLA⁺ STARD13⁺ microglia nodule in MS and (**h**) an HLA⁺ STARD13⁻ microglia nodule in stroke. **i** In MS, microglia nodules

more often express STARD13 compared to microglia nodules in stroke ($p < 0.05$). NAWM quantified in $n = 7$ MS donors and $n = 5$ stroke donors, NAWM adjacent to nodules quantified in $n = 7$ MS donors and $n = 6$ stroke donors, nodules quantified in $n = 7$ MS donors and $n = 6$ stroke donors. **j** An HLA⁺ IAH1⁺ microglia nodule in MS and (**k**) an HLA⁺ IAH1⁻ microglia nodule in stroke. **l** Microglia nodules in MS do not significantly more often express IAH1 compared to those in stroke. NAWM quantified in $n = 7$ MS donors and $n = 6$ stroke donors, NAWM adjacent to nodules quantified in $n = 7$ MS donors and $n = 6$ stroke donors, nodules quantified in $n = 7$ MS donors and $n = 6$ stroke donors Bar plots show mean value ± standard deviation. Significance was tested with a two-sided quasibinomial generalized linear model without correction for multiple testing, *$p$ value $< 0.05$, **$p$ value $< 0.01$, ***$p$ value $< 0.001$. Bar plots show mean value ± standard deviation. Source data are provided as a Source Data file.

progression. Furthermore, clinical progression is associated with more factors than new lesion formation, such as atrophy and ongoing axonal damage[44].

Gene expression was deconvoluted for cell types through integration of a dataset obtained by Absinta et al.[41]. Nodule tissue in MS compared to NAWM contained a higher proportion of microglia, a lower proportion of oligodendrocytes, and a higher proportion of endothelial cells. The lower proportion of oligodendrocytes may indicate demyelination of axons encapsulated by nodules. Although

IHC showed no difference in percentage of microglia nodules in contact with vessels in MS compared to stroke, the inflammatory milieu of the vessel in MS may differ from that in stroke. In MS, vessels are preferential lesion formation sites[45], and previously, we have shown that in the NAWM, there is an increase in perivascular tissue-resident memory T cells, which may be influencing the milieu in which microglia nodules in MS reside[46]. Therefore, the cell type composition of nodule tissue in MS suggests that microglia nodules may represent the first stage of MS lesion formation. Future studies focusing on single-

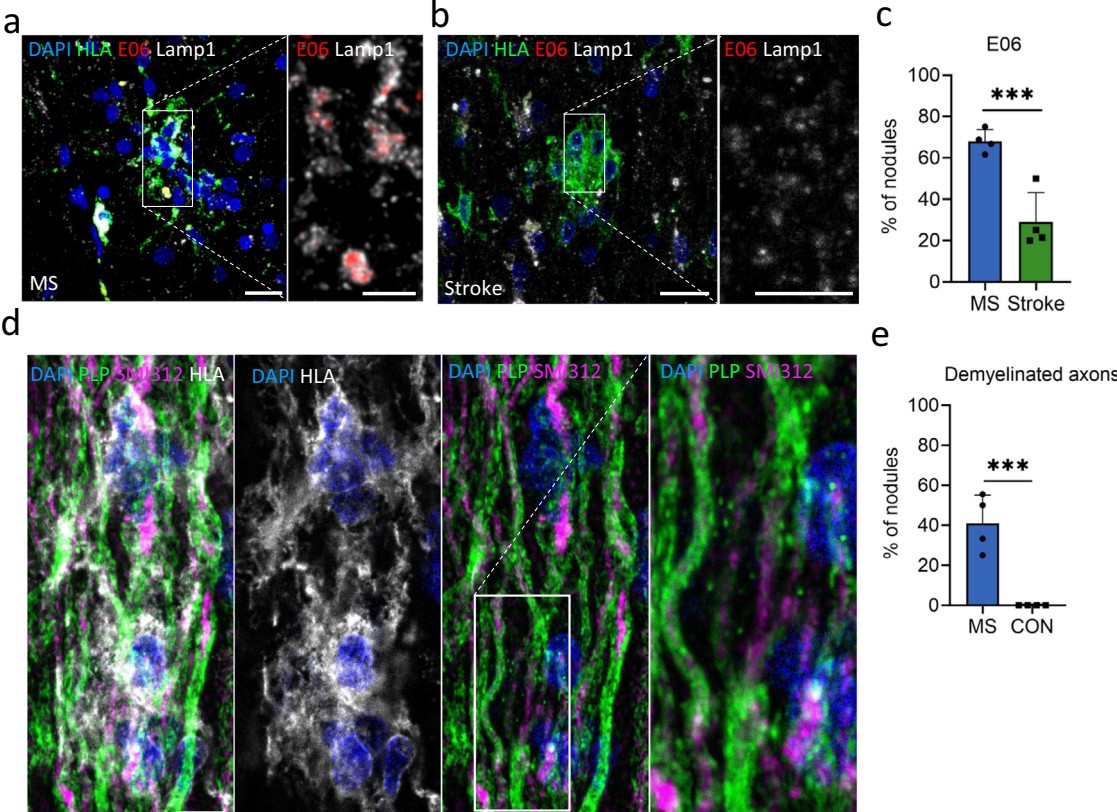

**Fig. 6 | Microglia nodules in MS are involved in demyelination.** STED microscopy of IHC stainings. **a** An HLA⁺ microglia nodule (in green) in MS containing lysosomal (Lamp1, in gray) oxidized phospholipids (E06, in red) and (**b**) an HLA⁺ microglia nodule (in green) in stroke that does not show oxidized phospholipids (E06, in red) in the lysosomes (Lamp1, in gray). Scale bars indicate 20 μm in overview images, and 10 μm in zoomed images. **c** Microglia nodules in MS have more often phagocytosed oxidized phospholipids compared to microglia nodules in stroke. Quantified in $n = 4$ MS donors and $n = 4$ stroke donors ($p = 3.2e−12$). Bar plot shows mean value ± standard deviation. Significance was tested with a two-sided quasibinomial generalized linear model without correction for multiple testing, ***$p$ value < 0.001. **d** In MS NAWM in the optic nerve, partially demyelinated axons (SMI312 in magenta, PLP in green) encapsulated by microglial nodules (HLA in gray) were found, which was not seen in control optic nerve WM, which (**e**) was significantly different ($p = 2.2^{−16}$). Quantified in $n = 4$ MS donors and $n = 4$ control donors. Scale bars indicate 20 μm. Source data are provided as a Source Data file.

cell sequencing and spatial transcriptomics will gain further insight on cell-specific gene expression.

Microglia nodules highly expressed genes that were previously implicated in MS pathology, such as *LPL*, *CXCL16*, *CD14*, *CDKN1A*, and *CHI3L1*[10,35]. Moreover, *CXCL16*, *MX1*, *HLA-DRB5*, *ISG15*, *IFNAR2*, and *IL1B* were previously found upregulated in active lesions or the rim of mixed active/inactive lesions, and *CXCL16* and *CHI3L1*, related to lipid binding, were found upregulated in perilesional areas of active lesions and are indicative for the expansion of lesions[27,28,32]. *CXCL16*, *CHI3L1*, *IFNAR2*, and *HLA-DRB5* have furthermore been suggested as potential prognostic markers in MS[33,36–38]. These genes of interest were not differentially expressed in the non-nodular NAWM in MS compared to the non-nodular WM in stroke. From this, we conclude that microglia nodules in MS show signs of lesion-associated microglia activation and are not part of a diffuse reaction of chronic damage in MS NAWM.

Interestingly, we found MS microglia nodule-specific upregulation of genes associated with lipid metabolism and catabolism (*DAGLB*, *IAH1*, *PLCD3*, *FABP5*, *ACLY*, *CHI3L2*, *STARD13*, and *GPCPD1*)[47]. On the protein level, microglia in the non-nodular (NA)WM rarely expressed FABP5, DAGLB, IAH1, or STARD13, and microglia nodules in MS compared to in stroke were more often FABP5⁺, DAGLB⁺, or STARD13⁺. Of interest, microglia surrounding microglia nodules also rarely expressed FABP5, DAGLB, IAH1, or STARD13. This indicates that upregulation of lipid metabolism is not an intrinsic effect of MS nor a diffuse effect. Therefore, microglia nodules are likely activated by a driver within the microenvironment of the nodule itself. The percentage of IAH1⁺

microglia nodules was only numerically higher in MS compared to stroke, likely indicating that the increased expression of IAH1 is also driven by other cell types than by microglia alone. We hypothesize that lipid metabolic processes are key in progression of an MS nodule to an inflammatory demyelinating lesion. Previously, we have shown that there are more mitochondria in axons in the NAWM[43] that may precede lipid oxidation in MS[42]. Potentially, this is an important trigger in the formation of microglia nodules. Therefore, microglia nodules in MS and in stroke were stained for lysosomal oxidized phospholipids. Indeed, in MS, a higher percentage of microglia nodules contained phagocytosed oxidized lipids, indicating that potentially these microglia nodules had formed to clear up damaged myelin. Interestingly, in normal-appearing optic nerve tissue of MS donors, some axons encapsulated by microglia nodules were partially demyelinated, which was not found in control cases. It cannot be excluded that in stroke, axons encapsulated by microglia nodules are partially demyelinated. However, considering the gene expression analysis and IHC of lipid metabolism markers, it is likely that this partial demyelination is MS microglia nodule specific. Considering the differential gene expression and the partially demyelinated axons in the nodules in MS, some nodules seem to be involved in demyelination and the formation of new lesions.

In contrast to what was previously found[24], with gene expression analysis and IHC we show the presence of C1q associated with microglia nodules in both MS and stroke. As complement deposition is necessary for Wallerian degeneration[48], and Wallerian degeneration is

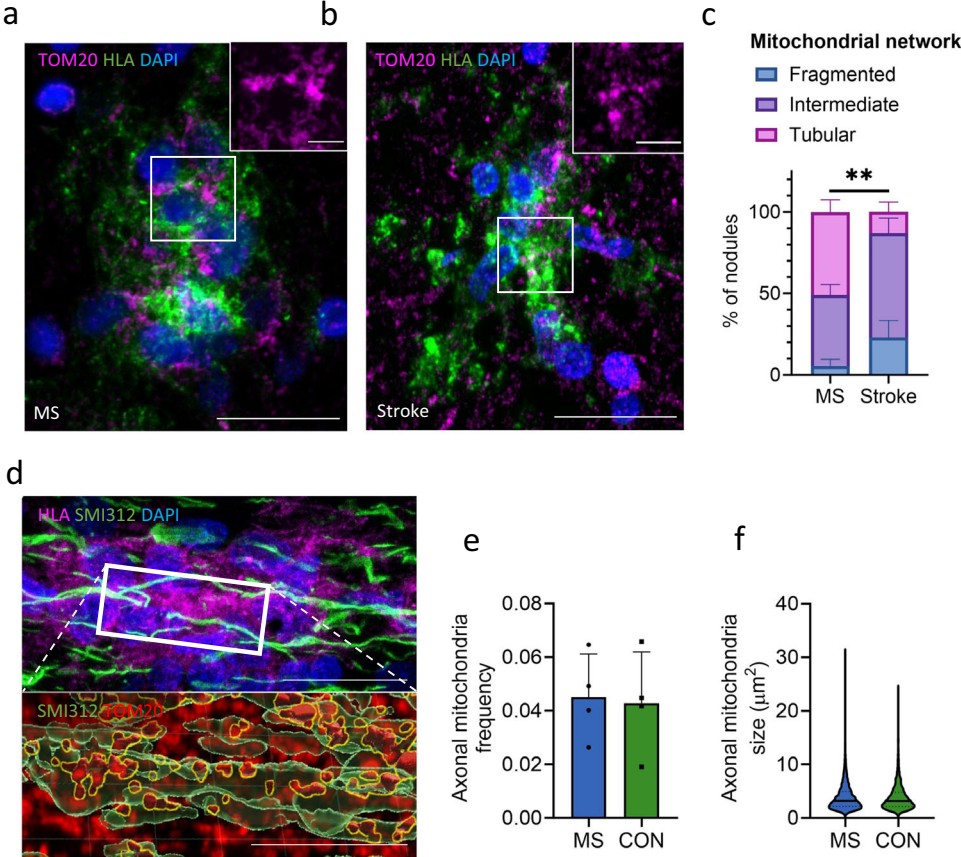

**Fig. 7 | Microglia nodules in MS show a more tubular mitochondrial network.** STED microscopy of IHC staining. Quantified in *n* = 4 MS donors and *n* = 4 stroke donors. **a** An HLA⁺ microglia nodule (in green) in MS with a fused and tubular TOM20 mitochondrial network (in magenta) and (**b**) an HLA⁺ microglia nodule (in green) in stroke with a fragmented TOM20 mitochondrial network (in magenta). Scale bars indicates 60 μm. **c** Microglia nodules in MS more often showed a tubular and fused mitochondrial network compared to microglia nodules in stroke (*p* = 0.009). Bar plot showing mean percentage per type of mitochondrial network in MS and in stroke. **d** Axonal mitochondria (TOM20 in red) in axons (SMI312 in green) encapsulated by a microglia nodule (HLA in magenta). In the animated panel (below), axons are reconstructed in green and mitochondria inside axons are highlighted with yellow outlining. Scale bar indicates 60 μm in overview images, and 30 μm in zoomed images. There was no difference in (**e**) frequency and (**f**) size of axonal mitochondria in axons encapsulated by HLA+ microglia in MS and controls. Bar plots show mean ± standard deviation. Violin plots show median and quartiles. Significance was tested with a two-sided quasibinomial linear model without correction for multiple testing, **\*\****p* value < 0.01. Source data are provided as a Source Data file.

a commonality between MS and stroke, microglia nodules in MS and stroke were likely to possess similarities.

Gene expression analysis indicates close proximity of activated lymphocytes that are influencing and being influenced by the microglia nodules in MS and not in stroke (*NCKAP1L, CASP3, JAK3, TCIRG1, CORO1A, GRB2, IRF8, TLR2, IL18*). With IHC, we found CD20⁺ B cells, CD138⁺ plasma cells, and IgG⁺ plasmablasts in close proximity to microglia nodules in MS and not in stroke, and in MS more microglia nodules were associated with activated, proliferating CD3⁺ T cells and CD38⁺ plasmablasts. Only a subset of the microglia nodules were in direct contact with lymphocytes, however lymphocytes can be involved in demyelination and acute neurodegeneration through secretion of soluble factors that can activate microglia and do not need direct cell contact[49,50]. Previously, Van Noort et al.[18] have hypothesized that microglia nodules in MS no longer reflect a local neuroprotective and reparative response if there is presence of lymphocytes. Possibly, the presence of activated T cells and Ig-producing B-cell blasts near some microglia nodules together with the phagocytosis of oxidized lipids in MS creates a volatile situation. This might indicate a critical turning point in which the nodule is not able to resolve and progresses into a demyelinating and inflammatory site. Previously, we showed MS NAWM to be enriched for perivascular B cells and T cells[46,51]. These lymphocytes may produce soluble factors contributing to lesion formation as cytokines and Igs[31,49,50]. Therefore, as we show lymphocytes

in close proximity to microglia nodules in MS, they may be contributing to the inflammatory environment in which microglia nodules reside.

Moreover, we found IgG⁺ plasmablasts and IgG deposition in the lumen of blood vessels, possibly IgG⁺ serum, in MS and not in stroke. We found that tissue containing microglia nodules in MS had significant upregulated Ig genes (*IGKC, IGHG1, IGHG2,* and *IGKV3-15*) compared to tissue containing stroke nodules. This corresponds with the characteristic high prevalence of intrathecal unique oligoclonal IgG production in MS and the presence of B cells in MS NAWM[50]. The absence of IgG in stroke donors is unsurprising. A recent study has shown that only few patients with acute ischemic stroke have intrathecal immunoglobulin synthesis (5.7%), of which 33% had a comorbid chronic inflammatory disease, such as MS[52]. Furthermore, most donors had experienced stroke >9 months after death, therefore the chance of still having ongoing IgG production is low in our cohort[53]. Although MS donors with any signs of stroke or ischemia were excluded from the cohort, microvascular pathology was not specifically investigated throughout the CNS, and we can therefore not fully exclude the effects of microvascular pathology. Previously, we have shown that microglia in the NAWM are immunosuppressed[54], and that Igs can break this immune tolerance of microglia cells through Fcγ receptors and thereby potentiate inflammation by microglia[55]. Therefore, the presence of Ig-producing cells near microglia nodules in MS in

combination with the activation of microglia through pro-inflammatory cytokines secreted by nearby, activated lymphocytes may represent a hazardous situation. Furthermore, complement deposition was found in microglia nodules in both stroke as well as in MS, but only in MS, Igs were found. This may lead to activation of the complement cascade and subsequently to MAC formation and cell death in MS and not in stroke[56]. Previously, it was shown that microglia nodules, both in stroke as well as in MS, encapsulate C3d[+] complement deposits associated with the degeneration of axons[17,24,29]. Here, we demonstrate that microglia nodules in MS are more often associated with C3d[+] complement deposits compared to microglia nodules in stroke, and microglia nodules in MS were also more often associated with MAC formation. MAC staining was found mainly in the cytoplasm of HLA[+] microglia cells, which could indicate osmolysis of the microglia nodule itself or phagocytosis of MAC-targeted cells by the microglia. This may be a key mechanism underlying progression of microglia nodules in MS to Ig-associated, complement-dependent, demyelinating inflammatory lesions. These findings suggest that MS therapies associated with loss of intrathecal oligoclonal bands may also associate with a reduced odds of nodules progressing toward a lesion. Likewise, a higher intrathecal oligoclonal Ig production at the first symptoms of MS is associated with higher odds of new lesion development on MRI and a swifter occurrence of relapses[57–59].

Our data indicate that microglia nodules in MS have a disturbed cell metabolism (*PPRC1*, *SAMM50*, *MTG1*, *ACOX1*, *COA4*) and are responding to as well as producing ROS (*ASPDH*, *BID*, *CRYZL1*, *IDH1*, *MAOB*, *SMAD3*). As axonal mitochondrial dysfunction occurs in many disorders, mitochondria may act as a central sensor for axonal degenerative stimuli. It is hypothesized that degradative processes are activated when concerted stimuli surpass the mitochondrial homeostatic capacity[60]. Surprisingly, the mitochondria frequency and size in axons encapsulated by microglia nodules were not affected in MS compared to stroke. Possibly, changes to the axonal mitochondria in the axons encapsulated by the microglia nodules are subtle, and electron microscopy may provide more insight in early morphological changes of axonal mitochondria[61]. Correction for the proportion of microglia in the MS nodules negated significance, however this is not very surprising as microglia are main producers of ROS and therefore, the proportion of microglia cells may play a role in the neurodegenerative properties of an MS nodule. Generally, activated microglia cells switch from oxidative phosphorylation to glycolysis and form a fragmented mitochondria network[62]. Notably, microglia nodules in MS generally possess a more tubular and less fragmented mitochondrial network compared to microglia nodules in stroke. We hypothesize that the combined activation of microglia cells by surrounding lymphocytes together with the phagocytosis of oxidized lipids may result in a hypermetabolic and hyperinflammatory state, as previously shown for atherosclerosis[63,64], in which the microglia rely both on glycolysis as well as oxidative phosphorylation. This can result in prolonged longevity and increased production of cytokines and ROS. We further show that microglia nodules in MS compared to stroke have upregulated genes associated with production of cytokines, specifically IFNγ and TNF (*IRF8*, *GADD*, *IFNAR2*, *CD84*, *LRRK2*, *FADD*, *LPL*, *TLR2*, *EGR1*, and *IL18*). Future studies need to elucidate a causal relation between the combined stimulation of the microglia cells through pro-inflammatory cytokines and phagocytosis of oxidized phospholipids, the shift in mitochondrial network, and the production of cytokines, as this opens up a potentially interesting therapeutic avenue.

In summary, we propose that some microglia nodules in MS that have the potential to progress into inflammatory and demyelinating MS lesions, whereas those in stroke will not. Therefore, differences between microglia nodules in MS and stroke can provide insight in mechanisms behind MS lesion formation. Here, we demonstrate that microglia nodules in MS upregulate lesion-associated genes and genes indicative of demyelination. We also show that some microglia nodules

in MS encapsulate partially demyelinated axons. Moreover, we here describe that a combination of activated T cells, Ig-producing B cells, and oxidized lipids in and around MS microglia nodules may together enable microglia nodules to become hypermetabolic, form MACs, and give rise to 'mini' MS lesions. Together, we conclude that some microglia nodules in MS are likely sites of lesion initiation, and represent an interesting therapeutic target to prevent early demyelination and MS lesion formation.

## Methods

### Characterization of MS lesions
The procedure for brain donation to the Netherlands Brain Bank (NBB) and the use of clinical and pathological information for research has been approved by the medical ethics committee of the VU medical center (Amsterdam, The Netherlands), approval number 2009/148. Donors provided informed consent for brain autopsy and for the use of material and clinical data for research purposes in compliance with national ethical guidelines. There were no discrepancies in self-reported sex and assigned sex.

Of 167 MS brain donors of the Netherlands Brain Bank MS cohort (NBB-MS, www.brainbank.nl), diagnoses were confirmed by a neuro-pathologist, and MS lesions were characterized as described previously by Luchetti and colleagues[5]. All tissue blocks (on average $23.5 \pm 9.7$ (standard deviation)) that were dissected during autopsy upon MRI or macroscopical appearance of lesions were stained for HLA-DR/proteolipid protein (PLP) to assess the MS lesion type[5,65]. The proportions of active, mixed active/inactive (mixed), inactive, and remyelinated lesions were calculated. Active lesions were defined by partial loss of PLP myelin staining and presence of HLA-DR[+] cells throughout the lesion. Mixed lesions were defined by an inactive demyelinated center with absence of PLP staining and HLA-DR[+] cells present at the border of the lesion. Microglia in active and mixed active/inactive lesions are stratified as ramified (score 0), rounded (score 0.5) or foamy (score 1). The microglia/macrophage activity score (MMAS) of each donor was calculated by dividing the sum of the scores of the microglia/macrophages values by the amount of active and mixed active/inactive lesions. Inactive lesions were defined by an inactive demyelinated center with no HLA-DR[+] cells throughout. Remyelinated lesions were defined by partially myelinated axons with similar numbers of HLA-DR[+] cells compared to the adjacent NAWM. In addition, we determined lesion load in brainstem tissue blocks as these are dissected at standard locations, which allowed us to compare the same brain region among donors[5]. A load of reactive sites was defined as regions of accumulations of HLA-DR[+] microglia cells in normal-appearing brainstem tissue, which are typically larger regions than nodules in which the HLA-DR[+] microglia cells do not need to be in contact with each other to be considered as such, and total lesion load was defined as all active, mixed, inactive, and remyelinated but not reactive sites in the brainstem tissue. Additionally, each donor was scored for yes or no presence of microglia nodules in any tissue block dissected.

### Post-mortem brain tissue selection for LDM and IHC
Frozen and mirror paraffin-embedded tissue from NAWM of MS ($n = 7$) and WM of stroke brain donors ($n = 7$) with HLA-DR[+] microglia nodules were matched for age, sex, post-mortem delay, and pH of the cerebrospinal fluid (CSF) (Table 1). The normal appearance of tissue was confirmed by intact PLP myelin staining, and microglia nodules were determined as minimally 4 HLA-DR[+] accumulating microglia. MS and stroke samples containing foamy microglia, lesions, or reactive sites were excluded. MS brain donors who had experienced any brain infarct during life were excluded. The WM tissue of stroke brain donors did not contain infarcts. Cryo-protected frozen optic nerve samples of MS ($n = 4$) and control donors ($n = 4$) were previously obtained[43].

## Laser dissection microscopy and RNA isolation

Frozen tissue sections (20 μm) were mounted on PARM MembraneSlides (P.A.L.M. Microlaser Technologies, Bernried, Germany) and dried for 48 hours at room temperature in a sealed box containing silica gel. Sections were fixed in ice-cold dehydrated acetone on ice for 10 min and dried at RT in a slide box with silica gel. Sections were incubated with biotinylated HLA (1:100) and RNase inhibitor (1:500) in PBS + 0.5% Triton X-100 for 15 min and ABC (1:800) with RNase inhibitor (1:500) in PBS for 15 min. Immunostaining was visualized with 3'3-diaminobenzidine (1:100; Dako) incubation for 5 min at RT, followed by counterstaining in cresylviolet (0.1% in 70% EtOH) for 10 seconds. A series of 70%–86%–100% ethanol was dripped over the section and sections were transferred to the laser dissection microscope (LDM) (ZEISS) immediately.

Of each donor, 90-151 microglia nodules and an equal amount of non-nodular (NA)WM tissue were collected from 8-22 sections, depending on the nodule frequency. Tissue was lysed in 50 μl Trisure (Bioline, London, UK), and RNA was isolated with the RNeasy Micro Kit (Qiagen) using an adapted protocol. Chloroform was added 1:5, and samples were vortexed and incubated on ice for 5 min. After centrifugation at 11,000 rpm for 15 min at 4 ˚C, the aqueous phase was transferred to a new tube. The remaining sample was incubated with 1 volume of chloroform, vortexed, and centrifuged at 11,000 rpm for 15 min at 4 ˚C. The aqueous phases were combined. 1 volume of 70% EtOH was added and the sample was transferred to the loading column. The sample was run through the column 3 times by centrifugation for 20 sec at 10,000 rpm at 4 ˚C. RW1 buffer was added to the column and centrifuged at 10,000 rpm for 20 sec at 4 ˚C. DNAse1 and RDD buffer were incubated for 15 min RT. The sample was washed with RPE buffer followed by 70% EtOH for 20 sec at 10,000 rpm, and the column was allowed to dry for 5 min at 14,000 rpm. 14 μl of RNase-free $H_2O$ was added to the column, incubated at RT for 2 min, and RNA was collected by centrifugation for 1 min at 14,000 rpm.

## RNA sequencing and gene expression analysis

PolyA-enriched mRNA sequencing on an Illumina NovaSeq6000 system and sequence alignment were performed by GenomeScan (Leiden, The Netherlands). Putative adapter sequences were removed from the reads when the bases matched a sequence in the TruSeq adapter sequence set using cutadapt (v2.10). Trimmed reads were mapped to the human reference genome GRCh37.75 using HiSAT2 v-2-1.0[66]. Gene level counts were obtained using HTSeq (v0.11.0)[67]. Statistical analyses were performed using the edgeR[68] and limma/voom[69] R/Bioconductor packages (R: v4.0.0; Bioconductor: v3.11). Seven highly abundant mitochondrial genes were removed from the dataset. Genes with more than 2 reads in at least 4 of the samples were retained. Count data were transformed to log2-counts per million (logCPM), normalized by applying the trimmed mean of M-values method[70], and precision weighted using voom[71]. One stroke donor (nodule and non-nodular WM sample) and one MS non-nodular NAWM sample were identified as outliers and removed from the dataset (Suppl Table 1). Differential expression was assessed using an empirical Bayes moderated t-test within limma's linear model framework including the precision weights estimated by voom and the consensus correlation between samples of the same donor (function 'duplicateCorrelation', limma package). The differential expression analysis was performed both with and without a covariate for the estimated microglia content (in percent). The proportion of microglia was determined using cell type deconvolution with dtangle[72] using the set of markers from Darmanis et al.[73] and using the script DeconvAnalysis.Rmd by Patrick et al.[74] as a template. To test for the possible presence of lymphocytes, we also performed cell type deconvolution using lymphocyte markers as reported by Schirmer et al.[75] and Palmer et al.[76]. Resulting p values were corrected for multiple testing using the Benjamini-Hochberg false discovery rate. Genes were re-annotated using biomaRt using the Ensembl genome databases (v103). All differentially expressed genes with an adjusted p value of <0.05 were sorted on logFC, and up to 50 top differentially expressed genes were summarized with the *p* value and logFC adjusted for microglia proportion indicated. Principal component analysis (PCA) was performed on the logCPM values of the 500 most variable genes to distinguish sources of variation. Functional annotation and gene ontology (GO) analysis was performed on genes significantly differentially expressed with adjusted $p < 0.05$ between groups using DAVID[47], with Homo sapiens as background set. As GO terms were utilized to find genes of interest, no multiple testing correction was performed.

## Immunohistochemistry

For IHC, (NA)WM tissue of MS ($n = 9$) and stroke ($n = 8$) brain donors was cut from frozen (20 μm) or paraffin-embedded (8 μm) tissue blocks not containing any lesions or reactive sites. For SMI312 and PLP, normal-appearing optic nerve tissue of MS donors was fixed overnight in 4% paraformaldehyde, protected in 30% sucrose for 24 hours, frozen, and cut at 20 μm[43]. For paraffin-embedded tissue, antigen retrieval was performed as indicated in Table 3 for 10 min in a microwave at 700 W. Sections were incubated overnight at 4 °C with primary antibodies (Table 3) diluted in incubation buffer (for paraffin tissue: 0.5% gelatin and 0.5% Triton X-100 in TBS; for frozen tissue: 1% bovine serum albumin and 0.5% Triton X-100 in phosphate-buffered saline, pH7.6). HLA-DR- and PLP-stained sections were incubated with HRP-labelled anti-mouse antibody (K5007, Dako Real EnVision detection system; Dako, Santa Clara, California, USA) for 1 hour, and immunostaining was visualized with 3'3-diaminobenzidine (DAB) (1:100, Dako) incubation for 10 min, followed by counterstaining with haematoxylin for 30 sec and mounting in Entellan (Merck, Kenilworth, NJ, USA). For CD38/HLA- and HLA/von Willebrand factor (VWF)-stained sections, immunostaining of HLA and VWF respectively was visualized with DAB (1:100, Dako) supplemented with 3% nickel, and CD38 and HLA, respectively, were incubated with avidin-biotin complex–alkaline phosphatase kit (Vector, Olean, NY, USA) (1:800) for 1 hour, and staining was visualized with the ImmPACT Vector Red Alkaline Phosphatase Substrate Kit (Vector). For fluorescent stainings, the sections were either incubated with a compatible fluorophore (1:400), or the staining was enhanced with a compatible biotinylated secondary antibody (1:400) for 1 hour followed by avidin-biotin complex – HRP kit (1:800) for 45 min, biotinylated tyramide (1:10,000) for 10 min, ABC (1:800) for 45 min at RT, and lastly, streptavidin-conjugated fluorophore (1:800) for 1 hour. All fluorescent stainings were incubated with Hoechst (1:1,000) for 10 min, 0.1% Sudan Black in 70% ethanol for 10 min and mounted in Mowiol.

## Quantification of immunohistochemistry

HLA-DR+ microglia nodules were visualized with an Axioplan2 microscope (Zeiss, Oberkochen, Germany). The entire tissue section was scanned to manually count nodule numbers in each section and corrected for size of the tissue section. To determine the size of microglia nodules, a picture was made of each nodule and a macro to automatically determine nodule size was developed using Image-Pro software (MediaCybernetics, Bethesda, MD, USA). An outline of each nodule was drawn manually, and an area mask was placed to capture HLA-DR-stained microglia nodules using a greyscale intensity threshold >50 for HLA-DR/PLP stainings and a threshold >110 for HLA-DR stainings. The total area of HLA-DR-stained microglia nodules was automatically calculated and expressed as μm².

For CD138, CD3, CD20, CD4, CD8, and PCNA, stainings were visualized using a confocal laser-scanning microscope (SP8; Leica, Wetzlar, Germany) with the software LASX, magnification 40x. Each tissue section was scanned for HLA-DR+ or IBA1+ microglia nodules, and pictures were made with each nodule present in the middle to detect immune cells around microglia nodules in a radius of 150-180 μm.

**Table 3 | Antibodies overview**

| Antigen | Supplier (cat#) | Clone | Dilution | Antigen retrieval |
|---|---|---|---|---|
| C1qB | Abcam (ab92508) | EPR2981 | 1:100[a] | Tris EDTA buffer pH9 |
| C3d | Dako (A0063) | Polyclonal | 1:300 | Citrate buffer pH6 |
| C5b9 | Dako (M077701-8) | aE11 | 1:100 | Citrate buffer pH6 |
| CD138 | BioRad (MCA2459T) | B-A38 | 1:250 | Citrate buffer pH6 |
| CD20 | Dako (M0755) | L26 | 1:100 | Citrate buffer pH6 |
| CD3 | Dako (A0452) | Polyclonal | 1:100 | Citrate buffer pH6 |
| CD38 | Atlas antibodies (HPA022132) | Polyclonal | 1:3000 | Citrate buffer pH6 |
| CD4 | Dako (M7310) | 4B12 | 1:100 | Tris EDTA buffer pH9 |
| CD8 | BD Biosciences (641400) | SK1 | 1:500 | Citrate buffer pH6 |
| DAGLB | Atlas Prestige (HPA069377) | Polyclonal | 1:50[a] | PBS pH7.6 |
| E06 | Avanti (330002 S) | T15 | 1:100 | PBS pH7.6 |
| FABP5 | RabMab (ab255276) | EPR22552-64 | 1:2000 | Tris EDTA buffer pH9 |
| HLA-DR-DP-DQ | Dako (M0775) | CR3/43 | 1:100 | Citrate buffer pH6 |
| IAH1 | Invitrogen (PA5-65270) | Polyclonal | 1:50[a] | PBS pH7.6 |
| Iba1 | Wako (019-19741) | Polyclonal | 1:500 | Citrate buffer pH6 |
| IGG | Abcam (ab218427) | Polyclonal | 1:100 | Citrate buffer pH6 |
| LAMP1 | Abcam (ab24170) | Polyclonal | 1:200 | Citrate buffer pH6 |
| MBP | Sigma (AB980) | Polyclonal | 1:200 | Citrate buffer pH6 |
| PCNA | Santa Cruz (sc-25280) | PC10 | 1:1000 | Citrate buffer pH6 |
| PLP | Secotec (MCA839G) | plpc1 | 1:3000 | Citrate buffer pH6 |
| SMI312 | Eurogentec (SMI-312R) | SMI-312R | 1:6000 | Citrate buffer pH6 |
| STARD13 | Invitrogen (PA5-63622) | Polyclonal | 1:300[a] | PBS pH7.6 |
| Tomm20 | RabMab (ab186735) | EPR15581-54 | 1:100[a] | Citrate buffer pH6 |
| VWF | Atlas antibodies (AMAb90928) | CL1950 | 1:500[a] | Citraconic anhydride pH6 |

[a]Staining enhanced.

Pictures were processed and analyzed using Fiji software[77]. For VWF, CD38, C1qB, C3d, MAC, FABP5, DAGLB, IAH1, and STARD13, scans were made on the Axio slide scanner, x20 magnification (ZEISS, Oberkochen, Germany). For VWF, the percentage of all nodules in each section in contact with a vessel was quantified. Of each section, all HLA-DR$^+$ microglia nodules were annotated as positive or negative for C1qBb, MAC, FABP5, DAGLB, IAH1, and STARD13, and for CD38 nodules were annotated as positive or negative for CD38$^+$ cells in a radius of 150-180 μm on Qupath (version 0.4.3). The percentage of HLA$^+$ cells in the non-nodular (NA)WM expressing DAGLB, FABP5, IAH1, and STARD13 was quantified using cell profiler in Qupath of on average 5,100 HLA$^+$ cells per donor per staining. Of microglia nodules with a CD138$^+$, CD3$^+$, CD20$^+$, or CD38$^+$ lymphocyte nearby, the percentage of those in direct contact were quantified and of those not in direct contact, the distance from the lymphocyte to the nearest ramification of the microglia nodule was measured. The percentage of non-nodular HLA$^+$ microglia adjacent to microglia nodules ( <100 μm distance) expressing DAGLB, FABP5, IAH1, and STARD13 was also quantified using cell profiler in Qupath. In MS, 16.4 ± 22.1 nodules were found per section, and in stroke, 6.1 ± 7.3 nodules were found per section. In MS, 13.5 ± 7.6 HLA$^+$ cells were detected per nodule adjacent to nodules, and in stroke, 14.1 ± 9.0 HLA$^+$ cells were detected per nodule adjacent to nodules.

SMI312–PLP–HLA triple staining, LAMP1–E06–HLA triple staining, Tomm20–HLA double staining, and Tomm20–HLA–SMI312 triple staining were visualized at 63x using stimulated emission depletion (STED) microscopy (STEDYCON; Abberior Instruments, Göttingen, Germany) for z-stacks. 3D-rendered images were analyzed for (partial) demyelination in Fiji. For PLP triple staining, E06 triple staining and Tomm20 double staining, z-stack images were taken of all HLA$^+$ microglia nodules in each section on the STED microscope at magnification 63x with 0.5 μm step size. For E06, using Fiji plugin for ImageJ, all microglia nodules were annotated as positive or negative for LAMP1$^+$ E06$^+$ phagocytosed oxidized phospholipids within the nodule. For Tomm20, the length of the mitochondria network inside microglia was measured for each nodule using Fiji. The mitochondrial network inside the nodule was considered as fragmented if all mitochondria were <2 μm, intermediate if the largest mitochondria was between 2–5 μm, and tubular if the largest mitochondria was >5 μm in size[78], and the axonal mitochondria size and frequency was measured with Imaris (v9.7; Bitplane, Zurich, Switzerland). For the SMI312–PLP–HLA triple staining, the number of nodules in MS and control optic nerve and the number of nodules encapsulating partially demyelinated axons were quantified.

## qPCR of genes involved in immunoglobulin production

Frozen tissue section (20 μm) was cut from stroke ($n = 6$) and MS ($n = 6$) (NA)WM tissue containing microglia nodules and lysed in 800 μl TRIsure. RNA was isolated according to manufacturer's instructions (Bioline). Briefly, chloroform (1:5) was added to each TRIsure sample and after centrifugation, the aqueous phase was collected, followed by incubation with 1 μg glycogen (Roche, Basel, Switzerland) for 30 min in ice-cold isopropanol at -20 °C. Precipitated RNA was washed in ice-cold 75% ethanol and diluted in 20 μl deionized water.

Synthesis of cDNA was performed according to manufacturer's instructions, using the Quantitect Reverse Transcription kit (Qiagen, Hilden, Germany). 50 ng RNA was mixed with 1 μl gDNA Wipe-out buffer and incubated for 2 min at 42 °C, followed by incubation with QuantiTect Buffer, RT Primer Mix, and Quantitect Reverse Transcriptase for 30 min at 42 °C and incubation for 3 min at 95 °C.

To determine gene expression of immunoglobulin (Ig) genes, quantitative polymerase chain reaction (qPCR) was performed. Control WM tissue and tissue collected from MS lesions and lymph node was used as negative and positive control samples, respectively. Primers were designed at the Integrated DNA Technologies website

(eu.ifdna.com). Optimal primers were selected based on dissociation curve and specificity, examined on cDNA derived from control, MS NAWM, and MS lesioned tissue. Gene expression was normalized on the mean of two housekeeping genes, *GAPDH* and *EEF1A1*. For each gene, the relative expression was calculated using the $2^{-\Delta\Delta CT}$ method. Primers used for reverse transcription (RT)-qPCR are provided in Suppl Table 2.

## Statistical analysis

Data obtained from IHC and RT-qPCR was tested for normality by a Shapiro–Wilk normality test, followed by parametric or non-parametric two-sided tests for numeric data, two-sided quasibinomial generalized linear mixed models for proportional data, or a two-sided $\chi^2$ test for binomial data to define p-values. Statistical analyses were performed in Rstudio (version 1.2.5033; Rstudio, Boston, MA, USA) for R (version 4.2.0), using key packages ggplot2, lme4, car, plyr, ggpubr, Hmisc, and corrplot. *P* values < 0.05 were considered significant.

## Reporting summary

Further information on research design is available in the Nature Portfolio Reporting Summary linked to this article.

## Data availability

The sequencing data generated in this study have been deposited in the Gene Expression Omnibus (GEO) database under accession code GSE234700. Source data are provided as a Source Data file. Source data are provided with this paper.

## Code availability

The code corresponding to the findings of this study is available from the corresponding author upon request.

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

## Acknowledgements

We are grateful to the brain donors and their families for their commitment to the Netherlands Brain Bank donor program. We thank Niamh McNamara for her input on the manuscript. Funding for this research was obtained from MS Research grant 19-1079 and MoveS grant 17-975, received by IH.

## Author contributions

A.M.R.v.d.B., M.v.d.P, J.S, I.H., and J.H. contributed to the conception and design of the study, A.M.R.v.d.B., M.v.d.P., and M.C.J.V. contributed to collection of sequencing samples, A.M.R.v.d.B., A.J., H.J.E., and P.D.M. contributed to the analysis of sequencing results, A.M.R.v.d.B., M.v.d.P., N.F., and A.M.B. contributed to acquisition and analysis of pathological data. All authors contributed to the interpretation of findings and drafting and editing of the manuscript.

## Competing interests

The authors declare no competing interests.
