## [Peer Review File · Nature Communications]

Profiling of microglia nodules in multiple sclerosis reveals propensity for lesion formationREVIEWER COMMENTS

Reviewer #1 (Remarks to the Author):

This is an interesting study by van den Bosch et al. investigating microglia nodules in multiple sclerosis (MS) by RNA-sequencing and high-resolution immunohistochemistry. They performed laser capture microscopy and compared findings in MS with data from stroke tissue samples. After bulk tissue RNA-sequencing, the authors performed cell type deconvolution to predict presence of cell types based on gene expression. In summary, the authors conclude that MS cases with presence of microglia nodules showed a stronger level pathology compared to cases without nodules. They found more MS lesion associated genes in nodule tissues from MS vs stroke.

In total, authors analyzed 90-151 microglia nodules vs 8-22 normal appearing white matter (NAWM) tissue areas, however, exact details about capturing areas are not provided. Hence, it is not clear if nodules were captured from lesion core vs rim or NAWM areas. Currently, it is also not clear what distance to lesion rims were chosen for laser capture. As the distance to the inflamed lesion rim would likely determine the level of inflammatory activity in those nodules, this information would help better understand the transcriptomic changes and differences between samples and compared to stroke. In that respect, laser capture microscopy is biased and does not provide a larger picture as through unsupervised spatial transcriptomics.

In general, this study is well performed and written and provides a rich resource into the transcriptomic changes happening in tissue areas characterized by a high density of microglia cells that the authors describe as microglia nodules. As the authors state, the presence of these microglial aggregates is not limited to MS but also appear in other neuroinflammatory conditions (primary and secondary) such as in stroke or Alzheimer's disease. In the discussion, the authors suggest that these microglia nodules represent an early event indicative for lesion formation, which is quite speculative. In general, someone would expect that microglia aggregates in a primary (autoimmune driven) inflammatory disease such as MS share more genes related to immune cell function and activation, which would not necessarily mean that these nodules represent a lesion starting point.

The results about mitochondrial/metabolic changes with respect to microglial nodules are quite speculative, however, it would be interesting to follow up if microglia in those aggregates exhibit an exhausted phenotype that drives them into a cellular state that makes myelin/lipid digestion difficult. Although this is a large dataset, single-nucleus and/or high-resolution spatial transcriptomics would help make current conclusions less speculative by precise cell type mapping and link microglial nodules to lesion and non-lesion areas. The current bioinformatic approach (using deconvolution based on known cell type specific marker genes) is a valid but not precise method to identify and map cell types in the tissue samples examined.

Reviewer #2 (Remarks to the Author):

The study deals with an interesting topic, the state of activation of microglia in microglia nodules in MS. The authors have first analyzed in a very large samples of MS autopsies the presence of microglia nodules in relation to clinical and pathological disease characteristics. This part in essence confirms that the presence of such nodules are associated with high lesion load and high load of active lesions. In this part microglia nodules were also compared between MS and stroke patients. There was no significant difference in the size of the nodules, but the nodule frequency was higher in MS. In the second part of the study these microglia nodules in MS and stroke were characterized by their gene expression patterns and by immunohistochemistry. This part is based on a smaller sample number, including the normal appearing white matter of 7 MS and 7 stroke patients. Overall, this part of the study shows that MG activation between MS and stroke differs

mainly in the pro-inflammatory activation. Many of the pro-inflammatory pathways, which are known from active MS lesions have been found here in the activated microglia from the microglia nodules of MS patients. Overall, the authors conclude that a subset of these nodules of activated microglia may represent initial lesions in MS.

It has been shown before in many studies that in MS cases, in particular in the vicinity of active lesions there is a pro-inflammatory microglia activation, which is very similar to that seen in the active lesion areas. This study describes the pathways of microglia activation in greater molecular detail, but does not identify a unique new pathway. Whether this pattern of activation is specific for microglia in the nodules is also not shown, since microglia in the nodules have not been compared with the microglia outside the nodules in the same section.

Despite this basic critique, the study is well and carefully performed and contains useful and new details.

There are, however, some points, which should be addressed:

- 1) The authors show demyelination in the microglia nodules in MS. Why did the authors not include stroke microglia nodules for this specific aspect? This would be important, since their conclusions are in part based on this observation.
- 2) The design of the experiments regarding spatial relation of the findings is not clear. The authors analyzed the tissue surrounding microglia nodules in a distance of 150 to 180 micrometers. This just tells you something about the global tissue environment. Much more knowledge would be obtained, when the study focuses on cells or expression patterns within the nodules or on structures in direct contact with nodular microglia.
- 3) Regarding immunoglobulins and MAC: On what cells are immunoglobulins deposited or bound. What structures are really labeled by C5-9; in the image it looks like the cytoplasm of a small cell, which is difficult to interpret regarding pathogenetic significance.
- 4) Overall, I miss the distinction between expression of molecules within microglia cells versus in neuronal or glial cells within the nodules. As an example, mitochondrial injury is rather expected within neurons and glia within the nodules that within microglia cells. This also applies for several other molecular markers described here.

Reviewer #3 (Remarks to the Author):

The authors proposed a combined neuropathological and molecular analysis of microglia nodules in post-mortem MS compared to stroke cases, suggesting enhanced expression of genes previously found to be upregulated in MS associated with lipid metabolism, presence of T and B cells, production of immunoglobulins and cytokines, activation of the complement cascade, and metabolic stress.

Even if this is an interesting pathological study, there are some major points, including first of all the lack of originality and novelty.

The Detection and classification of remyelinated lesion is not accurate, as some of the results indicate.

There is no assessment and analysis of the potential presence and characterization of vascular abnormalities that could have a key role in the comparison between MS and stroke.

The evidence of absence/presence of small caliber vessels in nodule area and in correlation with lymphocyte infiltration could help to better understand the origin of the infiltrating cells.

Point by point reply for manuscript ID #NCOMMS-23-29116-T: “Profiling of microglia nodules in multiple sclerosis reveals propensity for lesion formation”. Below, we give a point-by-point reply (R) to the reviewers’ comments (C), followed by any revisions that were made in the manuscript. In the manuscript, changes are highlighted as red text.

We thank the reviewers for the time invested in evaluating our manuscript, and appreciate their valuable insights and remarks.

Reviewer 1:

This is an interesting study by van den Bosch et al. investigating microglia nodules in multiple sclerosis (MS) by RNA-sequencing and high-resolution immunohistochemistry. They performed laser capture microscopy and compared findings in MS with data from stroke tissue samples. After bulk tissue RNA-sequencing, the authors performed cell type deconvolution to predict presence of cell types based on gene expression. In summary, the authors conclude that MS cases with presence of microglia nodules showed a stronger level pathology compared to cases without nodules. They found more MS lesion associated genes in nodule tissues from MS vs stroke.

R) We thank the reviewer for their kind words.

C1 & 2) In total, authors analyzed 90-151 microglia nodules vs 8-22 normal appearing white matter (NAWM) tissue areas, however, exact details about capturing areas are not provided. Hence, it is not clear if nodules were captured from lesion core vs rim or NAWM areas. Currently, it is also not clear what distance to lesion rims were chosen for laser capture. As the distance to the inflamed lesion rim would likely determine the level of inflammatory activity in those nodules, this information would help better understand the transcriptomic changes and differences between samples and compared to stroke. In that respect, laser capture microscopy is biased and does not provide a larger picture as through unsupervised spatial transcriptomics.

R1) The reviewer raised the concern that the capturing area was not clearly described in the manuscript text. All nodules as well as non-nodular (NA)WM was captured from tissue blocks that did not contain any lesion, reactive sites, or foamy microglia, as seen with the HLA-PLP staining. This is made more clear in the manuscript in lines 128.

R2) The reviewer highlights the potential of spatial transcriptomics to further understand the role of microglia nodules in MS. At this moment, we find that spatial sequencing platforms lack either the sequencing depth, chip size, or spatial resolution to find subtle but highly important differences between microglia nodules in MS and stroke, and between microglia nodules and surrounding (NA)WM in MS and stroke. We agree with the reviewer that spatial sequencing in time will be of high interest to gain further in-depth understanding of the role of microglia nodules in MS lesion formation, however considering these substantial technical difficulties, we do not consider this a feasible option at this moment in time. Our interest and suggestion for such research to be performed in the future has been added to the discussion in the manuscript in lines 503-505.

C3) In general, this study is well performed and written and provides a rich resource into the transcriptomic changes happening in tissue areas characterized by a high density of microglia cells that the authors describe as microglia nodules. As the authors state, the presence of these microglial aggregates is not limited to MS but also appear in other neuroinflammatory conditions (primary and secondary) such as in stroke or Alzheimer’s disease. In the discussion, the authors suggest that these microglia nodules represent an early event indicative for lesion formation, which is quite speculative.

In general, someone would expect that microglia aggregates in a primary (autoimmune driven) inflammatory disease such as MS share more genes related to immune cell function and activation, which would not necessarily mean that these nodules represent a lesion starting point.

R3) We agree with the reviewer that an inflammatory environment alone is not enough to initiate lesion formation in MS. Our data shows that the inflammatory environment may be an important trigger, and we hypothesize that additional co-activation of the microglia nodules with other stimulations such as complement or oxidized phospholipids will determine if the microglia nodule will resolve or progress into an MS lesion. In **Fig. 6**, the partial demyelination of axons encapsulated by microglia nodules indicates that a subset of these nodules are indeed involved in early lesion formation. To distinguish between an intrinsic MS response and an MS microglia nodule response, is very important to stratify between alterations to microglia found in relation to MS and those in relation to specifically microglia nodules in MS. In our study design, we have controlled for this on the transcriptional level, as we have not only compared microglia nodules in MS to microglia nodules in stroke, but we have also compared the non-nodular surrounding white matter in MS to stroke as well as microglia nodules to their respective surrounding tissue. In MS nodules, we saw an upregulation of MS lesion-associated genes, which was not seen in the non-nodular surrounding tissue nor in the stroke nodules. This has been highlighted in the manuscript in lines **512-513**.

To further investigate the specificity of our genes of interest, we have now additionally quantified the percentage of HLA⁺ cells in the surrounding (NA)WM tissue expressing proteins involved in lipid metabolism (DAGLB, FABP5, IAH1, STARD13) at the protein level. For each measurement, on average 5,100 HLA⁺ non-nodular microglia cells were quantified per donor on the same sections as the nodules were quantified. As can now be seen in **Figure 5c, f, i & j**, the percentage of HLA⁺ cells in the (NA)WM expressing DAGLB, FABP5, IAH1, or STARD13 is low, and no significant differences are observed between MS and stroke. Although we cannot definitively exclude the possibility of intrinsic alterations in the NAWM in MS due to the inflammatory nature of the disease, these additional quantifications make it more likely that we are studying a volatile phenotype of MS microglia nodules that makes it likely that a subset of these nodules progress into MS lesions. This can be found back in the manuscript in lines **229-231, 420-423, 518-519, 520**.

C4) The results about mitochondrial/metabolic changes with respect to microglial nodules are quite speculative, however, it would be interesting to follow up if microglia in those aggregates exhibit an exhausted phenotype that drives them into a cellular state that makes myelin/lipid digestion difficult. Although this is a large dataset, single-nucleus and/or high-resolution spatial transcriptomics would help make current conclusions less speculative by precise cell type mapping and link microglial nodules to lesion and non-lesion areas. The current bioinformatic approach (using deconvolution based on known cell type specific marker genes) is a valid but not precise method to identify and map cell types in the tissue samples examined.

R4) We agree with the reviewer that this is an interesting hypothesis, and we have mined our sequencing dataset for exhaustion markers as described by Bido *et al.*, *Nature Commun.* 12, 6237 (2021). We have studied the expression of *CD22*, *NOS1*, *CYBB*, *CD68*, and *CYBA*. Results can be found in Figure for the reviewers 1. These markers for microglia exhaustion are not significantly upregulated by microglia nodules in MS compared to the surrounding NAWM nor compared to stroke nodules, however the expression for these genes is numerically higher for all markers. As we do not have an appropriate control for exhausted microglia to compare the CPM with, we can unfortunately not confirm nor rule out the possibility that MS microglia nodules are exhausted, and a follow-up study should focus on this. We have added this to the manuscript in lines **343-345**.

Reviewer 2:

The study deals with an interesting topic, the state of activation of microglia in microglia nodules in MS. The authors have first analyzed in a very large samples of MS autopsies the presence of microglia nodules in relation to clinical and pathological disease characteristics. This part in essence confirms that the presence of such nodules are associated with high lesion load and high load of active lesions. In this part microglia nodules were also compared between MS and stroke patients. There was no significant difference in the size of the nodules, but the nodule frequency was higher in MS. In the second part of the study these microglia nodules in MS and stroke were characterized by their gene expression patterns and by immunohistochemistry. This part is based on a smaller sample number, including the normal appearing white matter of 7 MS and 7 stroke patients. Overall, this part of the study shows that MG activation between MS and stroke differs mainly in the pro-inflammatory activation. Many of the pro-inflammatory pathways, which are known from active MS lesions have been found here in the activated microglia from the microglia nodules of MS patients. Overall, the authors conclude that a subset of these nodules of activated microglia may represent initial lesions in MS.

R) We thank the reviewer for their feedback on our manuscript and for their thorough and thoughtful review of our work.

C1) It has been shown before in many studies that in MS cases, in particular in the vicinity of active lesions there is a pro-inflammatory microglia activation, which is very similar to that seen in the active lesion areas. This study describes the pathways of microglia activation in greater molecular detail, but does not identify a unique new pathway. Whether this pattern of activation is specific for microglia in the nodules is also not shown, since microglia in the nodules have not been compared with the microglia outside the nodules in the same section. Despite this basic critique, the study is well and carefully performed and contains useful and new details.

R1) We agree with the reviewer that we have indeed not identified a unique, new pathway. However, we are convinced that this is not a weakness but a strength of our study. As microglia nodules in MS and not microglia nodules in stroke nor microglia in the non-nodular surrounding tissue of MS compared to stroke are involved in pathways previously shown associated with MS lesions, we show that likely a subset of these microglia nodules in MS are indeed contributing to lesion formation. We would like to emphasize that microglia outside the nodules have been compared on a transcriptomic level, as these are present in the tissue collected from the surrounding non-nodular (NA)WM. To substantiate this point, we have additionally quantified the percentage of HLA⁺ cells in the (NA)WM that express DAGLB, FABP5, IAH1, or STARD13 on a protein level. For each measurement, on average 5,100 HLA⁺ non-nodular microglia cells were quantified per donor on the same sections as the nodules were quantified. As can now be seen in Figure 5c, f, i & j, the percentage of HLA⁺ cells in the (NA)WM expressing DAGLB, FABP5, IAH1, or STARD13 is low, and no significant differences are observed between MS and stroke. This has been highlighted and added to the manuscript in lines 229-231, 420-423, 518-519, 520.

There are, however, some points, which should be addressed:

C2) The authors show demyelination in the microglia nodules in MS. Why did the authors not include stroke microglia nodules for this specific aspect? This would be important, since their conclusions are in part based on this observation.

R2) Prior to the start of this project, a lot of care went into the harmonisation of tissue processing, such as handling and dissemination of post-mortem tissue, preparation of the tissue for

immunohistochemistry and laser microdissection, RNA isolation and sequencing. This allowed to control for false-positive errors due to technical differences. Another project was ongoing simultaneously, and here we were performing electron microscopy imaging and high-resolution immunohistochemistry of sub-cellular components of MS and non-demented control optic nerve tissue, possible due to a special dissection protocol in place at that time at the Netherlands Brain Bank. As the samples were not collected initially for the nodules project, stroke samples were not obtained. In the optic nerve samples collected, we encountered many nodules in the MS tissue and sporadic nodules in the control tissue. As the special dissection protocol enabled high-resolution imaging and in the optic nerve all axons are aligned in the same direction, we were able to study the myelin sheets of individual axons encapsulated by nodules. This also enabled us to answer another important comment of the reviewer **(R/C5)**. We agree with the reviewer that in hindsight, obtaining stroke tissue for high resolution imaging would be optimal, however unfortunately, this is technically no longer possible in a fully harmonised way. We have commented on this caveat more elaborately in the manuscript in lines **530-533**.

C3) The design of the experiments regarding spatial relation of the findings is not clear. The authors analyzed the tissue surrounding microglia nodules in a distance of 150 to 180 micrometers. This just tells you something about the global tissue environment. Much more knowledge would be obtained, when the study focuses on cells or expression patterns within the nodules or on structures in direct contact with nodular microglia.

R3) We concur with the reviewer's comment on the importance of the spatial relation. With the exception of lymphocytes, all immunohistochemistry was specifically focussed on structures or expression patterns that are either expressed by the microglia in the nodules themselves or by structures encapsulated by microglia in the nodules. Previous studies have found T and B cells distant from the sites of initial myelin and axonal injury, and hypothesize that although they are therefore not likely involved in direct cell contact-mediated active demyelination, they are likely involved in demyelination and acute neurodegeneration through soluble factors, which diffuse into the tissue and trigger demyelination through microglia activation (Machado-Santos *et al.*, Brain 141, 2066, 2018 & Bogers *et al.*, eBioMedicine 89, 104465, 2023). Therefore, in order for T and B cells to be influencing the microglia nodules, they do not necessarily need to be in direct cell to cell contact.

To gain a deeper spatial understanding of the localisation of the lymphocytes in relation to nodules, we quantified the percentage of nodules with lymphocytes nearby that were in direct contact with the nodule, and for those not in direct contact, we quantified the distance from the lymphocyte to the closest ramification of the nodule. Of all nodules with CD3⁺ T cells in their proximity, 14% were in direct contact with CD3⁺ T cells, and in the remaining 86%, the CD3⁺ T cells were found at an average distance of $50.47 \pm 35 \mu\text{m}$. Of all nodules with CD20⁺ B cells in their proximity, none were in direct contact, and B cells were found at an average distance of $81.52 \pm 53.36 \mu\text{m}$. Of all nodules with CD138⁺ plasma cells, 8% were in direct contact with CD138⁺ plasma cells, and in the remaining 92%, the CD138⁺ plasma cells were found at an average distance of $71.48 \pm 39.44 \mu\text{m}$. Of all nodules with CD38⁺ plasmablasts in their proximity, 38% were in direct contact with CD38⁺ plasmablasts, and in the remaining 62% the CD38⁺ plasmablasts were found at an average distance of $71.48 \pm 39.44 \mu\text{m}$. This data has been added to the manuscript in lines **199-203, 231-234, 382-386, 546-549**.

C4) Regarding immunoglobulins and MAC: On what cells are immunoglobulins deposited or bound. What structures are really labeled by C5-9; in the image it looks like the cytoplasm of a small cell, which is difficult to interpret regarding pathogenetic significance.

R4) We agree with the reviewer that the location of immunoglobulin deposition or secretion can be of high importance. Therefore, we have stained the tissue for immunoglobulin G heavy chain. In MS,

4/7 donors showed IgG⁺ staining. Interestingly, all four of these MS donors showed staining in the lumen of some vessels, where sometimes a nodule was nearby, and in two MS donors, IgG⁺ cells, likely plasmablasts, were found in proximity to nodules. This has been added to **Figure 3g-h** and in the manuscript to lines 386-392, 544, 558.

Regarding the C5b-9 staining, as the reviewer notes, the staining is mainly found in the cytoplasm of HLA⁺ cells. This could indicate osmolysis of the microglia nodule itself, or phagocytosis of C5b-9 targeted cells by the microglia. This has been added to the manuscript in lines 574-575.

C5) Overall, I miss the distinction between expression of molecules within microglia cells versus in neuronal or glial cells within the nodules. As an example, mitochondrial injury is rather expected within neurons and glia within the nodules that within microglia cells. This also applies for several other molecular markers described here.

R5) We acknowledge the reviewer's comments, however with bulk sequencing it is not possible to trace the expression back to the cell type, such as with single-cell sequencing or spatial transcriptomics. Bulk sequencing of laser micro-dissected tissue enabled the sequencing of samples with enrichment of microglia nodules, without compromising on sequencing depth. Therefore, we were able to pick up subtle differences between microglia nodules in MS and in stroke, and with their respective (NA)WM. With immunohistochemistry, we have validated that the expression of our genes of interest are likely coming from the microglia cells in the nodule, however we cannot fully exclude the possibility of other cell types driving some of the expression data. This is more comprehensively written in the manuscript in lines 495-496.

We agree with the reviewer that the genes of interest indicating mitochondrial injury and stress can also be a product of the axons encapsulated by the nodule. Therefore, we have performed a triple staining for HLA, TOMM20, and SMI312 on the optic nerve samples disseminated for high-resolution immunohistochemistry discussed in **C/R2** to visualise the mitochondrial network in axons, which are encapsulated by nodules. As visualised in **Figure 7d-f**, of 4 MS and 4 control donors, axonal mitochondria in axons encapsulated by nodules were imaged with STED microscopy, and images were processed and mitochondria were quantified as previously described (van den Bosch *et al.*, *Anna Neurol.* 93, 856, 2023). Interestingly, no difference was found between the axonal mitochondria frequency nor the axonal mitochondria size, so there is no indication of mitochondrial fragmentation, fusion, or mitophagy in axons encapsulated by nodules. This has been added to the manuscript in lines 235-236 245-246, 451-452, 457-458, 584-586.

Reviewer 3:

The authors proposed a combined neuropathological and molecular analysis of microglia nodules in post-mortem MS compared to stroke cases, suggesting enhanced expression of genes previously found to be upregulated in MS associated with lipid metabolism, presence of T and B cells, production of immunoglobulins and cytokines, activation of the complement cascade, and metabolic stress.

Even if this is an interesting pathological study, there are some major points, including first of all the lack of originality and novelty.

R) We thank the reviewer for their critical assessment of our work.

To our knowledge, this is the first work providing an integrated multi-modal analysis including a discovery driven genome wide RNA sequencing approach of the mechanisms described by the reviewer in the context of nodules in MS with stroke as controls, and we provide a highly detailed

and comprehensive dissection of the MS nodule as putative precursors of lesion formation in MS. In this study, we have extensively investigated and quantified donor-related characteristics of MS pathology, and mRNA expression and proteins in relation to lipid metabolism, activation of the complement cascade, and mitochondrial changes in the inflammatory environment of microglia nodules, providing insight in the formation of MS lesions. The combination of these approaches has provided insight in the formation of MS lesions and found potential therapeutic targets to resolve microglia nodules. As we have not included any tissue samples containing any MS lesions, our finding of MS-lesion associated genes expressed by nodules in MS and not in stroke is remarkable and highly suggestive for early MS lesion formation. Our study now provides an in-depth understanding of the microglia nodule and its role in MS lesion formation.

We therefore strongly believe that our work is an important addition to the already existing literature, and we have emphasised this in the manuscript in lines 472-474.

C1) The Detection and classification of remyelinated lesion is not accurate, as some of the results indicate.

R1) We would like to point the reviewer to manuscript lines 111-112, where the classification of remyelinated lesions has been provided which is according to earlier studies by us and others [1–5]. The detection of the remyelinated lesions per donor was done in a previous study [4].

1. van den Bosch A, Fransen N, Mason M, Rozemuller AJ, Teunissen C, Smolders J, Huitinga I (2022) Neurofilament Light Chain Levels in Multiple Sclerosis Correlate With Lesions Containing Foamy Macrophages and With Acute Axonal Damage. *Neurol Neuroimmunol Neuroinflamm*. 2022 Mar 3;9(3):e1154. doi: 10.1212/NXI.0000000000001154.
2. Kornek B, Storch MK, Weissert R, Wallstroem E, Stefferl A, Olsson T, Lington C, Schmidbauer M, Lassmann H (2000) Multiple sclerosis and chronic autoimmune encephalomyelitis: A comparative quantitative study of axonal injury in active, inactive, and remyelinated lesions. *Am J Pathol* 157:267–276. doi: 10.1016/S0002-9440(10)64537-3
3. Kuhlmann T, Ludwin S, Prat A, Antel J, Brück W, Lassmann H (2017) An updated histological classification system for multiple sclerosis lesions. *Acta Neuropathol* 133:13–24. doi: 10.1007/s00401-016-1653-y
4. Luchetti S, Fransen NL, van Eden CG, Ramaglia V, Mason M, Huitinga I (2018) Progressive multiple sclerosis patients show substantial lesion activity that correlates with clinical disease severity and sex: a retrospective autopsy cohort analysis. *Acta Neuropathol* 135:511–528. doi: 10.1007/s00401-018-1818-y
5. Van Der Valk P, De Groot CJA (2000) Staging of multiple sclerosis (MS) lesions: Pathology of the time frame of MS. *Neuropathol Appl Neurobiol* 26:2–10. doi: 10.1046/j.1365-2990.2000.00217.x

C2) There is no assessment and analysis of the potential presence and characterization of vascular abnormalities that could have a key role in the comparison between MS and stroke. The evidence of absence/presence of small caliber vessels in nodule area and in correlation with lymphocyte infiltration could help to better understand the origin of the infiltrating cells.

R2) The reviewer raises the valid point that blood vessels may relate to nodule or lesion formation. Indeed, as visualised in **Figure 2b**, a higher proportion of vascular cells were found in MS nodule tissue compared to surrounding tissue. As lymphocytes in the (NA)WM are preferentially located in the peri-vascular space, microglia nodules surrounding vessels in MS may be more prone to lesion formation. This was already mentioned in the manuscript in lines 489-492. To further investigate a

potential difference in nodules associated with vessels in MS and stroke we have double stained the tissue with endothelial marker VWF and Iba1, and we have quantified the percentage of microglia nodules in contact with vessels. As visualized in **Suppl. Fig. 2** and in the manuscript in lines **314-315, 497-599**. There was no difference in the percentage of nodules in contact with vessels in MS compared to stroke (MS: 81 out of 229 nodules, per donor $37.5\% \pm 11.4\%$; Stroke: 9 out of 36 nodules, per donor $23.0\% \pm 16.3\%$). Therefore, we found that nodules in MS are not more likely to contact a vessel compared to stroke, however, what they may encounter at the vessel site may play a role in progression or resolution of the nodule into a lesion.

We have now additionally investigated the location of immunoglobulin deposition with immunohistochemistry, as shown in **Figure 3g-h** and added to the manuscript in lines **386-389, 544, 558**, and we found that sometimes in MS and never in stroke vessels stained positive for IgG. This further implies the vessel as a potentially preferential lesion formation site in MS where microglia nodules, immunoglobulins and lymphocytes can interact.

Additionally, we have investigated the correlation of the percentage of nodules in contact with vessels and those in proximity to lymphocytes (VWF with CD38: $p=0.59$, $r=0.17$; VWF with CD138: $p=0.77$, $r=-0.14$; VWF with CD20: $p=0.65$, $r=-0.21$; VWF with CD3: $p=0.85$, $r=0.09$). Therefore, it seems not necessary for the nodule to be near a vessel to encounter lymphocytes and for the initiation of lesion formation.

REVIEWER COMMENTS

Reviewer #1 (Remarks to the Author):

The authors have made important revisions to the manuscript as suggested, however, certain points have not been addressed in the revision or are still unclear.

For example, in Fig. 6d the two images do not show the same region of interest, hence it would be important to display additional images (same area) with less/more channels: DAPI, SMI312 and PLP (left) and DAPI, SMI312, PLP, HLA (right). Then, it would be possible to check whether microglia cells/nodules were indeed associated with partially demyelinated axonal segments. Also, I would like to see a statistic (contacts of microglial nodules with demyelinated axonal segments vs non-demyelinated segments) as this would help strengthen the conclusion about the presence of nodules and lesion initiation as suggested by the authors. Furthermore, this would help explain the axonal mitochondrial changes associated with the presence of microglial nodules.

I appreciate the additional immunohistochemistry for gene expression markers in NAWM as shown in Fig. 5. This helps validate the gene expression data and strengthens the conclusions about the relevance/specificity for MS lesion/nodule areas.

The authors have responded to my previous suggestion that spatial transcriptomic approaches would help strengthen and support the current but have not decided to perform any spatial transcriptomic. It would be actually nice to see some additional spatial RNA mapping to bridge/complement their bulk tissue RNA-seq data with the immunohistochemistry. I agree that a large unbiased spatial RNA-seq data set would represent an entire new study, however, showing additional RNA in situ hybridization for some marker genes either using a manual multiplex assay like RNAscope or a larger automatic workflow as through Vizgen or Xenium (focusing on some nodule positive vs negative areas) would be important and support the data.

Lastly, I like that the authors included more STED imaging data about the mitochondrial changes in axons in Fig. 7. I am actually surprised that the results between MS and controls were not different, as it is known from previous studies that mitochondrial changes (size, morphology) are indeed happening and can be seen in inflammatory demyelination. Hence, I would be more careful interpreting the results shown in Fig. 7 as the gold imaging standard to assess mitochondrial pathology would be electron microscopy, which would be needed to make more in-depth and precise conclusions.

Reviewer #2 (Remarks to the Author):

The authors have clarified in their revision of the manuscript many of the points raised by the reviewers. Overall, the manuscript contains highly interesting data. However, there are points, which remain unclear and others, which became apparent through the addition of new data into the manuscript. These points mainly concern the interpretation of the findings.

1) The authors argue that microglia nodules may be the initial step in lesion formation in multiple sclerosis. However, as it has been shown before - and is also shown by the authors in the present study - areas of the NAWM containing microglia nodules are frequent and widespread in patients with progressive MS. This is in contrast to the rare appearance of new focal white matter lesions in this stage of the disease. Thus, the presence of such microglia nodules may be rather reflect the slowly progressive diffuse damage in the NAWM than the actual formation of new white matter lesions.

2) One key point, raised in the previous critique, was that the authors did not perform a careful comparison of microglia in the nodules with the adjacent non nodular microglia in the same section. When microglia activation in the nodules is related to T- and B-cells in the tissue, which is present at some distance of the nodules, it is expected that the same activation is also seen in

the adjacent non-nodular microglia in the same section. If a significant difference exists in the activation patterns between such nodular and adjacent non-nodular microglia, this may indicate that there is an additional driver within the microenvironment of the nodules itself. To answer this question it is not sufficient to analyze microglia activation in areas of NAWM, which does not contain nodules.

3) The authors now describe an additional phenotype (CD22, NOS1, CYBB, CD68, and CYBA) of "microglia exhaustion". However, this term is completely misleading. In the respective study (Bido et al 2021) these cells are described as highly cytotoxic microglia with prominent expression of molecules involved in oxidative damage and blockade of the oxidative stress was neuroprotective in this experiment. The authors used the term "phagocytic exhaustion" to describe that the phagocytic activity of these cells was reduced.

4) It is not clear, what the authors mean with the following sentence: "we found IgG deposition in some MS cases and never in stroke in the lumen of some vessels (4/7 MS donors)". Do the authors argue that the vessel lumen in stroke patients does not contain IgG? Are there thrombotic vascular occlusions in MS vessels, but not in stroke vessels? Is there intraluminal immune complex deposition in the absence of complement activation?

Reviewer #3 (Remarks to the Author):

Many thanks to the authors for the detailed revisions.
The manuscript is now ready for publication.

We thank the reviewers for evaluating our manuscript again and appreciate their valuable insights and remarks. Below, we reply point-by-point (R) to the reviewers' comments (C), followed by an explanation of the revisions that we made in the manuscript. In the manuscript, changes are highlighted with yellow to easily differentiate them from the previous revisions.

REVIEWER COMMENTS

Reviewer #1 (Remarks to the Author):

The authors have made important revisions to the manuscript as suggested, however, certain points have not been addressed in the revision or are still unclear.

C1) For example, in Fig. 6d the two images do not show the same region of interest, hence it would be important to display additional images (same area) with less/more channels: DAPI, SMI312 and PLP (left) and DAPI, SMI312, PLP, HLA (right). Then, it would be possible to check whether microglia cells/nodules were indeed associated with partially demyelinated axonal segments. Also, I would like to see a statistic (contacts of microglial nodules with demyelinated axonal segments vs non-demyelinated segments) as this would help strengthen the conclusion about the presence of nodules and lesion initiation as suggested by the authors. Furthermore, this would help explain the axonal mitochondrial changes associated with the presence of microglial nodules.

R1) We have adjusted Figure 6 accordingly and have added statistics to the figure and to the manuscript. As noted in the manuscript in lines 257-258, 457-458, and 896, in the optic nerve of 4 MS donors, a total of 20 nodules were found. Of these nodules, 9 had encapsulated partially demyelinated axons, with an average of $40.97\% \pm 14.23\%$ of nodules per donor. In control donors, in the optic nerve of 4 control donors, 24 nodules were found, of which 0 had encapsulated partially demyelinated axons ($p=2.2e-16$).

C2) I appreciate the additional immunohistochemistry for gene expression markers in NAWM as shown in Fig. 5. This helps validate the gene expression data and strengthens the conclusions about the relevance/specificity for MS lesion/nodule areas.

R2) We thank the reviewer for the kind words.

C3) The authors have responded to my previous suggestion that spatial transcriptomic approaches would help strengthen and support the current but have not decided to perform any spatial transcriptomic. It would be actually nice to see some additional spatial RNA mapping to bridge/complement their bulk tissue RNA-seq data with the immunohistochemistry. I agree that a large unbiased spatial RNA-seq data set would represent an entire new study, however, showing additional RNA in situ hybridization for some marker genes either using a manual multiplex assay like RNAscope or a larger automatic workflow as through Vizgen or Xenium (focusing on some nodule positive vs negative areas) would be important and support the data.

R3) The reviewer has suggested performing in situ hybridization to validate our RNA sequencing data, in addition to the immunohistochemical validation we have already performed. We have carefully considered this recommendation, and while we value the constructive feedback, we would like to share our perspective on why we believe it may not be appropriate:

1. *Biological relevance of protein validation:*

We have already successfully conducted thorough quantitative immunohistochemistry to validate our findings at the protein level, the gold-standard. Using quantitative immunohistochemistry, we have provided convincing spatial and cellular validation. Given that protein expression is a more direct representation of functional relevance, we believe our existing validation adequately supports the robustness of our RNA sequencing dataset. We believe that additional validation of our findings with a third descriptive method will not significantly increase the strength of our findings and will therefore add limited value to the manuscript.

2. *Limited nodule availability for quantitative analysis:*

Unfortunately, we no longer have tissue from all the donors included in our study. Please note, we carefully curated the cohort of nodule donors for this study over a period of 2 years. This involved assessing each tissue block of each stroke and MS donor that had come to autopsy at the Netherlands Brain Bank for the presence of nodules and the absence of any other signs of inflammation, infarction, or tissue damage. Therefore, finding additional donors will be challenging and time-consuming, and further, we cannot predict the projected timeline for new, eligible donors coming to autopsy. Performing quantitative in situ hybridization on such a limited sample size would not yield results that would reach statistical significance. Therefore, in situ hybridization experiments would only yield representative images, that would not add any statistically-supported evidence beyond what we have already shown with immunohistochemistry analyses.

3. *Time constraints:*

Conducting in situ hybridization experiments is time-consuming. Incorporating additional experiments of this nature is beyond the scope of the currently revised-manuscript, as this would be a new study in itself. We believe the challenging nature of these experiments would significantly delay the manuscript's submission, for minimal added value.

For the revision of this manuscript, as suggested by reviewer 2, we have further added more spatial evidence for MS microglia nodules as lesion formation sites, by quantifying the percentage of HLA⁺ microglia adjacent to microglia nodules in MS and stroke that are also positive for the lipid metabolism markers DAGLB, FABP5, IAH1, and STARD13. The response to the comment of this reviewer can be found below, at reviewer 2 C2/R2 and this has been added to the manuscript in lines 239-244, 433, and 533-536, to Figure 5, and to Suppl. Figure 4.

Overall, we believe that our current immunohistochemical analysis provides a comprehensive and biologically relevant validation of our novel findings.

C4) Lastly, I like that the authors included more STED imaging data about the mitochondrial changes in axons in Fig. 7. I am actually surprised that the results between MS and controls were not different, as it is known from previous studies that mitochondrial changes (size, morphology) are indeed happening and can be seen in inflammatory demyelination. Hence, I would be more careful interpreting the results shown in Fig. 7 as the gold imaging standard to assess mitochondrial pathology would be electron microscopy, which would be needed to make more in-depth and precise conclusions.

R4) We agree with the reviewer that electron microscopy would be suitable to detect more subtle changes in axonal mitochondria morphology. We have rewritten our interpretation of the axonal mitochondria to contain more nuance in lines 606-612.

Reviewer #2 (Remarks to the Author):

The authors have clarified in their revision of the manuscript many of the points raised by the reviewers. Overall, the manuscript contains highly interesting data. However, there are points, which

remain unclear and others, which became apparent through the addition of new data into the manuscript. These points mainly concern the interpretation of the findings.

C1) The authors argue that microglia nodules may be the initial step in lesion formation in multiple sclerosis. However, as it has been shown before - and is also shown by the authors in the present study - areas of the NAWM containing microglia nodules are frequent and widespread in patients with progressive MS. This is in contrast to the rare appearance of new focal white matter lesions in this stage of the disease. Thus, the presence of such microglia nodules may be rather reflect the slowly progressive diffuse damage in the NAWM than the actual formation of new white matter lesions.

R1) We agree with the reviewer that, considering the age and progressive clinical phenotype of the donors, it is not likely that many microglia nodules in MS will progress into MS lesions¹. As previously stated by Noort *et al.*, resolution of microglia nodules is even more likely than progression into MS lesions². In this manuscript, we do not wish to imply that all microglia nodules will become MS lesions. However, the molecular and immunohistochemical evidence we provide in this study indicates that likely MS lesions originate from microglia nodules. Therefore, a greater understanding of microglia nodules offers the opportunity to better understand MS lesion formation. NAWM abnormalities can be identified by MRI and have been attributed to focal microglial activation in the absence of clear demyelination^{3,4} or axonal pathology⁵. Recently, in a longitudinal study, Elliott *et al.* found that NAWM abnormalities, identified by relative T2 hyperintensity and reduced normalized magnetization transfer ratio, were furthermore more pronounced at locations with a high likelihood of developing subsequent MS lesions, both in relapsing-remitting MS and progressive MS. NAWM abnormalities could be detected as early as 144 weeks prior to lesion formation⁶. We have emphasized this more extensively in the manuscript in lines 54-57 & 79-81, and have added more nuance by removing 'Therefore, we conclude that microglia nodules in MS are indeed involved in demyelination, and a subset of nodules are forming a 'mini lesion'' from the results and added 'some' to the manuscript in lines 637.

1. Cree, B. A. C. *et al.* Secondary Progressive Multiple Sclerosis: New Insights. *Neurology* **97**, 378–388 (2021).
2. van Noort, J. M. *et al.* Preactive multiple sclerosis lesions offer novel clues for neuroprotective therapeutic strategies. *CNS Neurol. Disord. Drug Targets* **10**, 68–81 (2011).
3. Miller, D. H., Johnson, G., Tofts, P. S., Macmanus, D. & McDonald, W. I. Precise relaxation time measurements of normal-appearing white matter in inflammatory central nervous system disease. *Magn. Reson. Med.* **11**, 331–336 (1989).
4. De Groot, C. J. A. *et al.* Post-mortem MRI-guided sampling of multiple sclerosis brain lesions: Increased yield of active demyelinating and (p)reactive lesions. *Brain* **124**, 1635–1645 (2001).
5. Moll, N. M. *et al.* Multiple sclerosis normal-appearing white matter: Pathology-imaging correlations. *Ann. Neurol.* **70**, 764–773 (2011).
6. Elliott, C. *et al.* Abnormalities in normal-appearing white matter from which multiple sclerosis lesions arise. *Brain Commun.* **3**, (2021).

C2) One key point, raised in the previous critique, was that the authors did not perform a careful comparison of microglia in the nodules with the adjacent non nodular microglia in the same section. When microglia activation in the nodules is related to T- and B-cells in the tissue, which is present at some distance of the nodules, it is be expected that the same activation is also seen in the adjacent non-nodular microglia in the same section. If a significant difference exists in the activation patterns between such nodular and adjacent non-nodular microglia, this may indicate that there is an additional driver within the microenvironment of the nodules itself. To answer this question it is not sufficient to analyze microglia activation in areas of NAWM, which does not contain nodules.

R2) We appreciate the valuable critique and agree with the reviewer that it is important to investigate the microglia adjacent to the microglia nodules. Indeed, T- and B-cells in the tissue can be at some distance, and for microglia nodules to be associated with lesion formation and adjacent microglia not, they need to have a distinct phenotype. We hypothesize that immunoglobulins and cytokines secreted by these nearby lymphocytes together with the phagocytosis of oxidized phospholipids plays a role in activating the microglia and in activation of lipid metabolism, giving them the specific microglia nodule expression. Therefore, we have performed additional quantification of our immunohistochemical stainings of lipid metabolism markers DAGLB, FABP5, IAH1, and STARD13. Each nodule was previously annotated, and the annotation was expanded by 100 μm . Within each expanded annotation, nuclei detection was performed, number HLA⁺ cells were quantified and number HLA⁺-DAGLB⁺/FABP5⁺/IAH1⁺/STARD13⁺ cells were quantified. From this, the percentage of HLA⁺ cells that HLA⁺-DAGLB⁺/FABP5⁺/IAH1⁺/STARD13⁺ was calculated. In a supplementary figure for the reviewer, a DAGLB⁺ microglia nodule is shown surrounded by HLA⁺ microglia that are DAGLB⁻ or DAGLB⁺. In MS, on average 16.4 ± 22.1 nodules were found per section and in stroke on average 6.1 ± 7.3 nodules were found per section. In MS, on average 13.5 ± 7.6 HLA⁺ cells were detected per nodule adjacent to nodules, and in stroke on average 14.1 ± 9.0 HLA⁺ cells were detected per nodule adjacent to nodules. There was no difference between MS and stroke in the percentage of HLA⁺ cells surrounding microglia nodules that were also DAGLB⁺, FABP5⁺, IAH1⁺ or STARD13⁺ (DAGLB MS: $12.0\% \pm 8.0\%$, stroke: $8.6\% \pm 2.6\%$; FABP5 MS: $13.2\% \pm 5.4\%$, stroke: $15.3\% \pm 10.3\%$; IAH1 MS: $8.5\% \pm 5.8\%$, stroke: $5.5\% \pm 6.2\%$; STARD13 MS: $15.8\% \pm 7.9\%$, stroke: $5.4\% \pm 5.7\%$). The percentage of double-positive cells adjacent to the microglia nodules were comparable to the percentage of double-positive cells in the rest of the section, further away from the microglia nodules. In conclusion, even though the cellular micro-environment of microglia adjacent to nodules and of microglia nodules is similar, they are differentially activated. This has been added to figure 5 and has been adjusted in the manuscript to lines 239-244, 433, and 533-536, to figure 5 and to supplementary figure 4.

C3) The authors now describe an additional phenotype (CD22, NOS1, CYBB, CD68, and CYBA) of “microglia exhaustion”. However, this term is completely misleading. In the respective study (Bido et al 2021) these cells are described as highly cytotoxic microglia with prominent expression of molecules involved in oxidative damage and blockade of the oxidative stress was neuroprotective in this experiment. The authors used the term “phagocytic exhaustion” to describe that the phagocytic activity of these cells was reduced.

R3) We have adjusted the term accordingly in the manuscript in lines 355.

C4) It is not clear, what the authors mean with the following sentence: “we found IgG deposition in some MS cases and never in stroke in the lumen of some vessels (4/7 MS donors)”. Do the authors argue that the vessel lumen in stroke patients does not contain IgG? Are there thrombotic vascular occlusions in MS vessels, but not in stroke vessels? Is there intraluminal immune complex deposition in the absence of complement activation?

R4) The reviewer raises some intriguing questions for which we do not have the definitive answers yet. A recent study has shown that only few patients with acute ischemic stroke have intrathecal immunoglobulin synthesis (5.7%), of which 33% had a comorbid chronic inflammatory disease, such as MS¹. In our cohort, we have excluded stroke donors with any signs of inflammatory disease, limiting the chance of having included donors with IgG synthesis. Secondly, most donors had experienced stroke >9 months after death, therefore the chance of still having ongoing IgG production is low in our cohort². Furthermore, as have analyzed nodules and non-nodular tissue in the white matter away from ischemic areas, it is also unlikely that there would be intravascular thrombotic events at the site of the tissue investigated, although this cannot be excluded. For our

MS cohort, any donors with any clinical or pathological signs of stroke or ischemia were excluded. However, effects of microvascular pathology and hypertension/ dyslipidaemia-associated vascular changes were not specifically analysed, and we can therefore not fully exclude that this has occurred in the MS brain. This has been added to the manuscript in lines 574, 577-584. We do not know if there is intraluminal immune complex deposition in the absence of complement activation, but we hope that our future project that is currently being initiated to intensively investigate immunoglobulin depositions and intrathecal immunoglobulin will shed more light on this.

1. Laichinger, K. *et al.* No evidence of oligoclonal bands, intrathecal immunoglobulin synthesis and B cell recruitment in acute ischemic stroke. *PLoS One* **18**, 1–10 (2023).
2. Bernstein, J. J. & Goldberg, W. J. Injury-related spinal cord astrocytes are immunoglobulin-positive (IgM and/or IgG) at different time periods in the regenerative process. *Brain Res.* **426**, 112–118 (1987).

Reviewer #3 (Remarks to the Author):

Many thanks to the authors for the detailed revisions.

The manuscript is now ready for publication.

We thank the reviewer.

REVIEWERS' COMMENTS

Reviewer #1 (Remarks to the Author):

No further comments.

Reviewer #2 (Remarks to the Author):

The authors have properly addressed all points raised by the reviewers. The manuscript is now fine.